# CLOSING THE DATA-EFFICIENCY GAP BETWEEN AUTOREGRESSIVE AND MASKED DIFFUSION LLMS

## ABSTRACT

Despite autoregressive large language models (arLLMs) being the current dominant paradigm in language modeling, effectively updating these models to incorporate new factual knowledge still remains difficult. They resist knowledge injection via fine-tuning due to inherent shortcomings such as the "reversal curse" — the challenge of answering questions that reverse the original information order in the training sample. Masked diffusion large language models (dLLMs) are rapidly emerging as a powerful alternative to the arLLM paradigm, with evidence of better data efficiency and free of the "reversal curse" in pre-training. However, it is unknown whether these advantages extend to the post-training phase, i.e. whether pre-trained dLLMs can easily acquire new knowledge through fine-tuning. On three diverse datasets, we fine-tune arLLMs and dLLMs, evaluating them with forward and backward style Question Answering (QA) to probe knowledge generalization and the reversal curse. Our results confirm that arLLMs critically rely on extensive data augmentation via paraphrases for QA generalization, and paraphrases are only effective when their information order matches the QA style. Conversely, dLLMs achieve high accuracies on both forward and backward QAs without paraphrases; adding paraphrases yields only marginal gains. Inspired by the dLLM's performance, we introduce a novel masked fine-tuning paradigm for knowledge injection into pre-trained arLLMs. This proposed method successfully and drastically improves the data efficiency of arLLM fine-tuning, effectively closing its performance gap with dLLMs. We further show that the masked fine-tuning paradigm of arLLMs can be extended to the supervised fine-tuning (SFT) of mathematical capability. Across two models and two datasets, our masked SFT outperforms regular SFT.

## 1 INTRODUCTION

Despite auto-regressive large language models (arLLMs) being the main contributors to the modern success of language modeling, studies have demonstrated the difficulty of injecting new knowledge into pre-trained arLLMs by fine-tuning on documents that are not in the pre-training dataset (Ovadia et al., 2023; Mecklenburg et al., 2024; Gekhman et al., 2024; Soudani et al., 2024; Zhao et al., 2025; Lampinen et al., 2025). Fine-tuned models typically generalize poorly to downstream tasks such as question-answering (QA). An example failure mode is the famous "*reversal curse*", that LLMs fail to answer the questions in the reversed order of the training text (Berglund et al., 2023). Fine-tuning on multiple rewrites (i.e., paraphrases) of the documents can mitigate generalization issues but still falls behind in-context learning based external memory systems like RAG (Ovadia et al., 2023; Mecklenburg et al., 2024). This pitfall of arLLMs is a major obstacle that limits current models from being flexible life-long learners via weight updates.

As alternatives to the auto-regressive models, several recent masked diffusion large language models (dLLMs) have been scaled up to be as capable as arLLMs on multiple downstream tasks, with additional advantages such as high-throughput decoding of multiple tokens simultaneously (Nie et al., 2025a;b; Ye et al., 2025). Instead of factorizing the joint sequence probability strictly left-to-right, dLLMs learn the factorization under many permutations of token positions, enabling them to condition on any subset of tokens and predict the remainder. Though such an objective is harder than learning autoregressive factorization (Kim et al., 2025), dLLMs do not suffer from "*reversal*

Figure 1: A schematic summary of the results. First row: autoregressive LLM requires paraphrases for generalizing knowledge in the fine-tuning text to QA tasks, and suffer from reversal curse (i.e. fail to answer backward questions). Second row: masked diffusion LLM can easily generalize fine-tuning text to QA tasks in both forward and backward styles. Third row: inspired by the masked diffusion LLM, we propose a masked fine-tuning paradigm, that closes the fine-tuning gap between autoregressive LLMs and masked diffusion LLMs.

*curse*"(Nie et al., 2025b), and can achieve lower validation loss (i.e. token cross-entropy loss) than arLLMs in data-constrained settings (Prabhudesai et al., 2025; Ni & the team, 2025). However, most of the dLLMs studies have focused on the properties in the pre-training phase. Little is known about whether dLLMs also have advantages in the post-training phase, such as knowledge injection by fine-tuning.

In this study, we use three datasets to compare the data efficiency and performance of knowledge injection by fine-tuning in arLLM and dLLM models. We refer to "data-efficiency" as how much fine-tuning data (including paraphrases) is required for a pre-trained model to generalize the knowledge in the data to downstream tasks such as QA. We introduce a novel masked fine-tuning paradigm for arLLMs that emulates a diffusion-style mask reconstruction loss. Across all datasets, dLLMs show a consistent data-efficiency advantage over arLLMs, and our masked fine-tuning largely closes this gap, bringing arLLMs to strong performance without relying on paraphrases. We further show that adapting demasking objection to supervised fine-tuning (SFT) of arLLM outperforms regular SFT on math tasks. More specifically, we show the following results:

- arLLMs heavily rely on paraphrases to successfully generalize fine-tuning text to downstream QA tasks; arLLMs fail on backward style questions, and only paraphrases that reverse the information order in the sentences can mitigate the reversal curse.

- dLLMs can achieve high accuracy in both forward and backward questions without paraphrases; adding paraphrases only marginally helps. This establishes the knowledge injection data efficiency of dLLMs in the post-training phase.

- We propose a masked fine-tuning paradigm that fine-tunes arLLMs in a "masked infilling" way by giving masked samples in the context with instructions to recover the mask, and set the unmasked sample as the supervised fine-tuning target. The novel method closes the performance gap between arLLMs and dLLMs fine-tuning: it allows arLLMs of different sizes and model families to achieve strong performance in both forward and backward questions without paraphrases.

- We extend the masked fine-tuning paradigm to SFT tasks, and show that across two arLLMs and two math datasets masked SFT outperforms regular SFT.

Taken together, these results show that dLLMs are more data-efficient than arLLMs during post-training. We further show that this advantage can be transferred to arLLMs via our masked fine-

tuning paradigm. Our findings suggest the possibility to post-train an LLM to adapt to the changing world using a small amount of new knowledge texts, which could help address challenges in keeping AI systems updated with changing environment.

## 2 BACKGROUND

### 2.1 KNOWLEDGE INJECTION BY FINE-TUNING AND REVERSAL CURSE

An AI system intended for real-world deployment should continually acquire and integrate new information to adapt to evolving environments. Though LLMs have been successful on numerous tasks, they struggle to incorporate new knowledge into their weights. At least two factors contribute to this challenge. First, fine-tuning on new tasks can induce catastrophic forgetting of previously learned capabilities (Luo et al., 2023; Wang et al., 2023; Zhai et al., 2023; Zhang & Wu, 2024; Chen et al., 2024; Ren et al., 2024). Second, fine-tuning primarily reshapes surface behavior (e.g., tone and format) without reliably integrating new factual knowledge (Ovadia et al., 2023; Mecklenburg et al., 2024; Gekhman et al., 2024; Soudani et al., 2024; Zhao et al., 2025; Lampinen et al., 2025).

A failure mode of learning knowledge in the text is the "*reversal curse*", that after learning statements of the form "A is B", the model does not generalize to its inverse form "B is A". The reversal curse has been observed across the training phases and models (Berglund et al., 2023; Allen-Zhu & Li, 2025; Lv et al., 2024; Lin et al., 2024; Guo et al., 2024; Golovneva et al., 2024b; Lu et al., 2024). Even strong commercial models like GPT-4 and GPT-4o show signs of the reversal curse (Berglund et al., 2023; Nie et al., 2025b). The cause of the reversal curse has been theoretically attributed to an inherent limitation of the autoregressive training objective (Zhu et al., 2024; Kitouni et al., 2024) (more discussion in Appendix A.6). Common approaches to mitigate the reversal curse in autoregressive models include: (i) augmenting the training set with paraphrases (Lu et al., 2024), which requires substantial computation to construct; (ii) augmenting the training set with reordered sequences (Guo et al., 2024; Golovneva et al., 2024b), which often violate natural-language grammar and degrade overall language modeling performance; and (iii) replacing causal attention with bidirectional attention (Lv et al., 2024), which struggles to retrieve long information (e.g., a person's description). In contrast, our masked fine-tuning paradigm for arLLMs addresses the reversal curse without constructing paraphrase augmentations or altering the autoregressive objective. With only adjustments to the fine-tuning query and keeping the original documents as optimization targets, our method attains high accuracy across all datasets we evaluated.

### 2.2 MASKED DIFFUSION LANGUAGE MODELS

Recently, dLLMs have emerged as a strong competitor to arLLMs (Sahoo et al., 2024; Nie et al., 2025b; Ye et al., 2025). Comparing to autoregressive models, dLLMs use encoder-only transformers to generate text by iteratively unmasking tokens via a reversed discrete diffusion process. The training objective is to minimize the mask reconstruction loss Nie et al. (2025b):

$$\mathcal{L}(\theta) = -\mathbb{E}_{t,\boldsymbol{x}_0,\boldsymbol{x}_t} \left[ \frac{1}{t} \sum_{\ell=1}^{L} \mathbb{I}[\boldsymbol{x}_t^\ell \in \mathrm{M}] \log p_\theta(\boldsymbol{x}_0^\ell | \boldsymbol{x}_t) \right], \tag{1}$$

$\boldsymbol{x_0}$ is an original sequence of length $L$ sampled from the training data. The masking process is governed by the mask ratio $t$, which is sampled uniformly, resulting in the corrupted sequence $\boldsymbol{x_t}$. The set $\mathrm{M}$ denotes the indices of the tokens that were masked by the forward process at ratio $t$. The $\ell$-th token is considered for the loss only if it was masked. Such a loss objective has been shown to be the negative evidence lower bound (ELBO) on the data likelihood (Shi et al., 2024).

Recent studies report dLLMs are more data-efficient than arLLMs. When the training data is scarce, dLLMs keep improving with repeated use of the data and surpass arLLMs on validation loss, while arLLMs saturate the validation loss or increase it due to overfitting (Prabhudesai et al., 2025; Ni & the team, 2025). Prabhudesai et al. (2025) further shows that the lower validation loss in dLLMs can generalize to downstream tasks like ARC-Easy, and attributes its data efficiency to random masks as implicit data augmentation. However, whether these advantages persist in new knowledge

acquisition during post-training—where the model needs to learn generalizable knowledge through small fine-tuning sets—remains unclear.

Note that the denoising objective and bi-directional attention of dLLMs is closely related to the design choices in earlier language models such as BERT and T5/Flan-T5 Devlin et al. (2019); Raffel et al. (2020). However, they are less popular choices for studying the post-training phase properties. BERT is not a generative chat models thus cannot achieve QA tasks; the complexity of the encoder-decoder design of T5 hinders its advancement to match other state-of-art LLMs. Thus, in this study, we choose to use state-of-the-art dLLMs and arLLMs to probe the post-training properties involving question-answering.

## 3 DATASETS AND EXPERIMENTAL SETUPS

We focus on assessing LLMs' ability to learn new knowledge through fine-tuning. More specifically, LLMs are fine-tuned on a set of documents that contain knowledge unknown to the base LLM, and evaluated by open-ended QA tasks. The correctness of an answer is evaluated by the ROUGE-1 score (Lin et al., 2024; Jiang et al., 2025) between the generated answer and the ground truth answer, which we report as "accuracy." It measures the proportion of the words in the ground truth answer that appear in the generated answer. To better demonstrate the generation quality, we also show examples of model responses in all the experiments in A.7.

In the first part of the study, we use three representative datasets. Two are existing synthetic datasets from previous studies on the reversal curse, and each accompanied with a paraphrase set as data augmentation; one is constructed using recent Wikipedia articles. Previous studies on the reversal curse rarely use realistic datasets. In this study, we include the Wiki dataset to probe the reversal curse in a setting that reflects knowledge injection in the real world. See examples of each dataset in Appendix A.3.

The *NameDescription* (Berglund et al., 2023) contains 60 statements of different fictitious individuals, 30 each of the form "[name] is [description]" (N2D) and "[description] is [name]" (D2N). Lin et al. (2024) extended the dataset with an open-ended QA testing set. For each type of statement, the QA set contains two types of questions: "What is the name related to a given description" and "What is the description of a given name". Depending on whether the question is aligned with the original statement, each question is classified as "forward" or "backward" question (e.g. N2D statement with "What is the description of a given name" type of question is a forward question). The dataset also contains a paraphrase set, in which each statement is rewritten into 30 different versions, but the order of [name] and [description] in the paraphrases is always preserved as in the original statement (either N2D or D2N).

The *Biography* dataset is proposed in Allen-Zhu & Li (2024; 2025). Since the original dataset is not publicly available, we used a subset of 100 samples from a replication (Zheng et al., 2025). Each sample is a 6-sentence paragraph about a fictitious individual, detailing their birth city, birthday, college, and job information. Note that the name only appears in the first sentence and is replaced with a pronoun in the following sentences; thus, questions about the name are considered backward questions. Each sample also includes a paraphrase set of 5 paraphrases; the paraphrases do not change the order of the sentences but only alter the wording while preserving the information. The testing QA set has both forward (i.e., asking for an attribute given the name) and backward styles (i.e., asking for the name given 3 attributes from the person) questions.

The *Wiki* dataset contains 94 Wikipedia articles, constructed according to the protocol described in Pan et al. (2025). We crawl the Wikipedia pages under the category "2025 by month", and then further filter out the pages that were created before the year 2025. This procedure ensures that these real-world events are recent enough that both dLLM and arLLM models have minimal knowledge about them, which is justified by the models' accuracy before fine-tuning (Table 1). For each wiki article, we use GPT-o3-mini to generate QA pairs in both forward and backward styles. By prompting GPT-o3-mini, we construct two different paraphrase sets: one that retains the information in place while only changing the wording (same-order paraphrases); the other also changes the order of information in the article (permute-order paraphrases). 10 paraphrases of each type are generated for every wiki article. All LLM generated QAs or paraphrases are cross-checked by human and

invalid entries are filtered out and replaced. More details on constructing the datasets are provided in Appendix A.3.

We chose the arLLMs Llama-3.1-8B-Instruct, Llama-3.2-3B-Instruct (Dubey et al., 2024), Qwen2.5-7B-Instruct, Qwen3-4B-Instruct-2507 Yang et al. (2025) and dLLM LLaDA-8B-Instruct (Nie et al., 2025b) to conduct the experiments. LLaDA-8B-Instruct is directly comparable to Llama-3.1-8B-Instruct as they have similar parameter size and comparable general capability benchmarks. Fine-tuning and evaluation configurations are provided in the Appendix A.4.

| | NameDescription | | | | Biography | | Wiki | |
|---|---|---|---|---|---|---|---|---|
| | N2D-fwd | N2D-bwd | D2N-fwd | D2N-bwd | Fwd | Bwd | Fwd | Bwd |
| Llada before fine-tuning | 0.030 | 0.000 | 0.028 | 0.000 | 0.030 | 0.000 | 0.210 | 0.156 |
| Llada w/o paraphrases | 0.873 | 0.913 | 0.864 | 0.790 | 0.892 | 0.696 | 0.908 | 0.778 |
| Llada w paraphrases | 0.967 | 0.994 | 0.994 | 0.973 | 0.991 | 0.857 | 0.900 | 0.785 |
| Llama 8B before fine-tuning | 0.072 | 0.000 | 0.054 | 0.000 | 0.001 | 0.000 | 0.164 | 0.127 |
| Llama 8B reverse training | 0.637 | 0.125 | 0.113 | 0.088 | 0.180 | 0.011 | 0.609 | 0.425 |
| Llama 8B w/o paraphrases | 0.374 | 0.000 | 0.017 | 0.027 | 0.121 | 0.002 | 0.377 | 0.282 |
| Llama 8B w paraphrases | 0.910 | 0.004 | 0.925 | 0.071 | 0.962 | 0.001 | 0.685 | 0.396 |
| Masked Llama 8B w/o paraphrases | 0.658 | 0.949 | 0.992 | 0.923 | 0.971 | 0.598 | 0.980 | 0.930 |
| Masked Llama 8B w paraphrases | 0.969 | 0.996 | 0.928 | 0.832 | 0.965 | 0.816 | 0.905 | 0.794 |
| Llama 3B before fine-tuning | 0.078 | 0.067 | 0.000 | 0.000 | 0.001 | 0.001 | 0.106 | 0.159 |
| Llama 3B reverse training | 0.391 | 0.079 | 0.042 | 0.025 | 0.216 | 0.003 | 0.384 | 0.285 |
| Llama 3B w/o paraphrases | 0.230 | 0.078 | 0.017 | 0.000 | 0.032 | 0.001 | 0.292 | 0.229 |
| Llama 3B w paraphrases | 0.951 | 0.040 | 0.967 | 0.025 | 0.988 | 0.001 | 0.622 | 0.334 |
| Masked Llama 3B w/o paraphrases | 0.887 | 0.932 | 0.992 | 0.933 | 0.967 | 0.738 | 0.970 | 0.908 |
| Masked Llama 3B w paraphrases | 0.982 | 0.928 | 1.000 | 0.992 | 0.964 | 0.809 | 0.855 | 0.810 |
| Qwen 7B before fine-tuning | 0.039 | 0.030 | 0.000 | 0.000 | 0.018 | 0.000 | 0.205 | 0.231 |
| Qwen 7B reverse training | 0.902 | 0.092 | 0.450 | 0.350 | 0.540 | 0.026 | 0.678 | 0.481 |
| Qwen 7B w/o paraphrases | 0.966 | 0.043 | 0.367 | 0.000 | 0.357 | 0.003 | 0.676 | 0.371 |
| Qwen 7B w paraphrases | 0.987 | 0.063 | 0.954 | 0.000 | 0.956 | 0.003 | 0.712 | 0.414 |
| Masked Qwen 7B w/o paraphrases | 0.870 | 0.897 | 0.967 | 0.929 | 0.984 | 0.754 | 0.934 | 0.896 |
| Masked Qwen 7B w paraphrases | 0.957 | 0.810 | 0.933 | 1.000 | 0.960 | 0.828 | 0.867 | 0.821 |
| Qwen 4B before fine-tuning | 0.026 | 0.032 | 0.000 | 0.000 | 0.058 | 0.001 | 0.237 | 0.243 |
| Qwen 4B reverse training | 0.259 | 0.083 | 0.008 | 0.008 | 0.140 | 0.008 | 0.470 | 0.437 |
| Qwen 4B w/o paraphrases | 0.137 | 0.081 | 0.000 | 0.000 | 0.103 | 0.001 | 0.509 | 0.353 |
| Qwen 4B w paraphrases | 0.962 | 0.038 | 0.975 | 0.000 | 0.994 | 0.003 | 0.675 | 0.418 |
| Masked Qwen 4B w/o paraphrases | 0.928 | 0.902 | 0.950 | 0.967 | 0.944 | 0.623 | 0.907 | 0.870 |
| Masked Qwen 4B w paraphrases | 0.975 | 0.816 | 1.000 | 0.975 | 0.967 | 0.806 | 0.847 | 0.822 |

Table 1: Fine-tuning performance of the dLLM and arLLMs across all three datasets. Model names are shortened for presentation (see section 3 for full model names). "Masked" denotes the masked fine-tuning paradigm. "Reverse training" denotes the entity-level reverse-training paradigm proposed in Golovneva et al. (2024b), which we use as a baseline control. For the Wiki dataset, we use the same-order paraphrase set. Pink indicates clear failure (accuracy below 50%); turquoise indicates clear success (accuracy above 90%).

## 4 ARLLM KNOWLEDGE INJECTION RELIES ON PARAPHRASES

We first show that knowledge injection by fine-tuning in arLLMs heavily relies on paraphrases. This is known in previous studies (Berglund et al., 2023; Allen-Zhu & Li, 2025; Lin et al., 2024; Guo et al., 2024; Golovneva et al., 2024b). We demonstrate this observation on three datasets to set baselines for comparison with dLLM and our novel paradigm in the following sections.

We fine-tune arLLMs on dataset samples using the pre-training format (i.e., without a chat template). Without paraphrases, backward accuracy on the NameDescription and Biography datasets is close to 0, while the forward accuracy of NameDescription N2D and Biography does not completely fail but is still poor (Table 1). Adding same-order paraphrases drastically raises forward accuracy close to 1, while backward accuracy remains close to 0. Paraphrases do not help backward accuracy in NameDescription and Biography datasets because the construction of these datasets does not change the semantic order of the sentences. The trend is similar in the Wiki dataset (Table 1 and 2). While

the same-order paraphrases significantly increase forward accuracy, they only mildly increase backward accuracy. Using permute-order paraphrases increases both forward and backward accuracy, and the gap between them is smaller. Note that, due to the naturalness of this dataset, pre–fine-tuning accuracies are not as close to zero as in the other datasets (Table 1 and 2); nonetheless, they are sufficiently low to demonstrate the effectiveness of fine-tuning.

These results suggest that, in arLLM fine-tuning, paraphrases significantly improve QA accuracy, but help backward questions only when the paraphrases change the information order in the original text to be more aligned with the backward style. Note that the accuracy difference between fine-tuning with paraphrases and without paraphrases is not due to different training steps; in both cases, we train the models with sufficiently large epoch numbers; the reported accuracy is from the best checkpoints during the training (Figure 2, Appendix 6).

## 5 DLLM KNOWLEDGE INJECTION

We investigate the data efficiency of dLLMs for knowledge injection by fine-tuning, focusing on whether they require paraphrases to successfully handle both forward and backward QAs. We follow the original pretraining protocol (Nie et al., 2025b) to fine-tune LLaDA-8B-Instruct on the dataset samples using the loss defined in Eq. 1. On three datasets, the accuracy difference between fine-tuning with and without paraphrases is much smaller in the dLLM than in the arLLM (Table 1): dLLM without paraphrases can already achieve decent and comparable accuracies on both forward and backward questions; fine-tuning with paraphrases can further increase the accuracy by a small amount. These results together suggest that dLLM has superior data efficiency and is free of the reversal curse in the post-training phase. By plotting the testing accuracy across the training steps (Figure 2), we observe that arLLM fine-tuned without paraphrases improves QA accuracy only in the beginning of training, then quickly decreases, indicating overfitting. The dLLM without paraphrases,

|  | Wiki | |
|---|---|---|
|  | Fwd | Bwd |
| Llama 8B before fine-tuning | 0.164 | 0.127 |
| Llama 8B w/o paraphrases | 0.377 | 0.282 |
| Llama 8B w same-order paraphrases | 0.685 | 0.396 |
| Llama 8B w permute-order paraphrases | 0.721 | 0.628 |
| Llama 3B before fine-tuning | 0.106 | 0.159 |
| Llama 3B w/o paraphrases | 0.292 | 0.229 |
| Llama 3B w same-order paraphrases | 0.622 | 0.334 |
| Llama 3B w permute-order paraphrases | 0.667 | 0.550 |
| Qwen 7B before fine-tuning | 0.205 | 0.231 |
| Qwen 7B w/o paraphrases | 0.676 | 0.371 |
| Qwen 7B w same-order paraphrases | 0.712 | 0.414 |
| Qwen 7B w permute-order paraphrases | 0.761 | 0.621 |
| Qwen 4B before fine-tuning | 0.237 | 0.243 |
| Qwen 4B w/o paraphrases | 0.509 | 0.353 |
| Qwen 4B w same-order paraphrases | 0.675 | 0.418 |
| Qwen 4B w permute-order paraphrases | 0.693 | 0.639 |

Table 2: Fine-tuning performance of arLLM on the Wiki dataset. Pink indicates clear failure (accuracy below 50%).

on the other hand, does not show signs of overfitting. This finding echoes what has been found in comparing arLLMs and dLLMs in the pre-training phase (Prabhudesai et al., 2025; Ni & the team, 2025).

One may expect that fine-tuning dLLM converges slower than arLLM, because learning any-order factorization requires seeing more than one way of the factorizations (i.e., samples masked in different ways) (Xue et al., 2025; Kim et al., 2025). However, we found that dLLM converges at least as fast as arLLM (Figure 2, Table 4, Appendix Table 5); in the Biography dataset, dLLM even converges faster than arLLM. This indicates that dLLM does not trade better data efficiency and performance for more training compute; it requires the same or less training compute and fewer training samples but achieves better downstream performance.

## 6 MASKED FINE-TUNING OF ARLLM

Inspired by the supremacy of dLLM in knowledge injection by fine-tuning, we attempt to adapt its advantages to arLLM. If an instruct arLLM is capable enough, one may prompt an arLLM to act like a dLLM. Specifically, given a masked document and an instruction to recover the original document, if the model has knowledge of the original document and the masked document contains sufficient cues to retrieve this knowledge, an instruct arLLM is supposed to respond with the correct original document. If the arLLM does not already have the knowledge of the original document, setting the ground truth document as the supervised fine-tuning target may implicitly teach the model

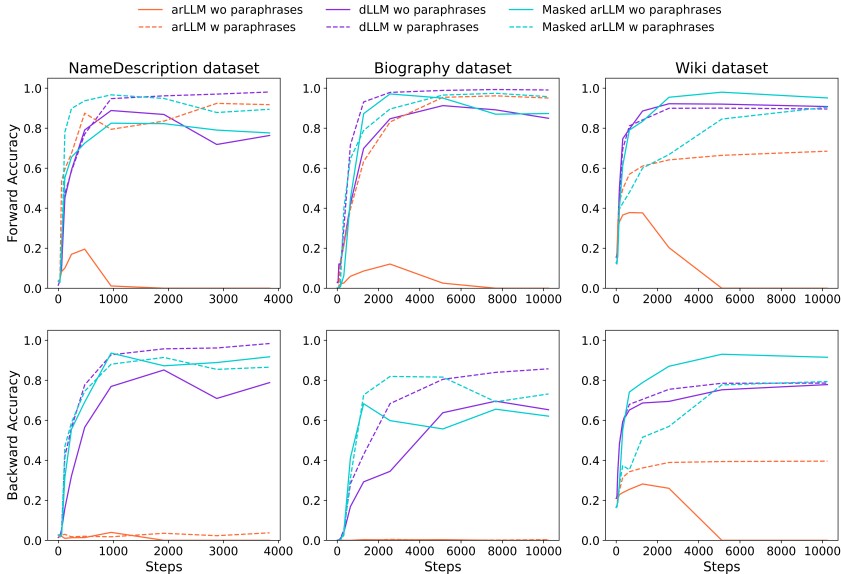

Figure 2: Training dynamics of arLLM (Llama 8B), dLLM (Llada), and masked arLLM (Llama 8B). For the NameDescription dataset, forward and backward accuracy are the average of N2D and D2N types. Paraphrases used in the Wiki dataset are the same-order paraphrases set. Due to the randomness of sampling the masks, we average across 4 random seed for the dLLM and masked arLLM on NameDescription and Biography Datasets. Curves for each seed are shown in Appendix Figure 10-11.

---

<|start_header_id|> user <|end_header_id|> \n\n [MASK] Barrington, known [MASK] and [MASK] for being [MASK] acclaimed director of the [MASK] reality masterpiece, "A [MASK] Through [MASK]." \n Return the recovered masked passage. <|eot_id|>
<|start_header_id|> assistant <|end_header_id|> \n\n Here is the recovered text:\ n

Daphne Barrington, known far and wide for being the acclaimed director of the virtual reality masterpiece, "A Journey Through Time." <|eot_id|>

---

Figure 3: An example of masked fine-tuning prompt. Random selection of text tokens are replaced by a [MASK] token. Highlighted tokens are used to compute the autoregressive loss.

that knowledge. We refer to this fine-tuning paradigm as "*masked fine-tuning*" of arLLM, and the resulting model as "*masked arLLM*". Masked fine-tuning of arLLM, from a broader perspective, establishes a training objective similar to that of dLLMs, wherein the model learns to reconstruct the unmasked sequence from a masked input. Following the dLLM noise sampling strategy, we randomly replace sample tokens with a reserved special token during training, where the mask ratio $t$ is sampled from a uniform distribution $U(0.05, 0.95)$. Note that each sample can be masked with a different mask ratio in different epochs. We evaluate the mask fine-tuned arLLM in the regular autoregressive way using the default chat template. The exact prompt used in the fine-tuning is provided in Figure 3 (more details in Appendix A.4), and the training objective formulation is:

$$\mathcal{L}(\theta) = -\frac{1}{\sum m_t} \sum_{t=1}^{T} m_t \, \log p_\theta\big(s_t \mid s_{<t}\big) \tag{2}$$

where $s$ is the constructed sequence which has the form of Figure 3; $m_t \in \{0,1\}$ selects which tokens are included in the loss which are 1 if the token is part of the original sample in the "assistant" window or it is an "end of sequence" token.

Overall, masked-finetuning of arLLM successfully inherits all the merits of the dLLM fine-tuning (Table 1, Figure 2). Masked arLLM surpasses arLLM fine-tuning in the pre-training style with a huge margin (Table 1) and achieves comparably high accuracy in both forward and backward question categories. Moreover, like dLLM, masked arLLM relies much less on paraphrases in the fine-tuning dataset to saturate the accuracy in most cases. The convergence rate of masked fine-tuning is also comparable to or in some cases faster than dLLMs (Figure 2, Table 4, Appendix Table 5), suggesting masked fine-tuning is data-efficient to achieve better down-stream QA tasks than traditional fine-tuning without costing additional training compute.

To confirm that the effectiveness of our masked fine-tuning stems from the objective itself and not merely a simple data augmentation effect (i.e., introducing varied input text prepended to the training sample), we conducted a control experiment, in which the masked document within the prompt was replaced entirely with random tokens (Appendix Figure 12). This substitution caused the accuracy of the masked fine-tuning to drop to the level of naive arLLM fine-tuning.

## 7 EFFECTS OF FINE-TUNING MASK RATIO

Previous studies (Allen-Zhu & Li, 2024; 2025) claim that bidirectional BERT-like models struggle with even forward style knowledge extraction due to the mask loss, which causes the model to learn incorrect associations between tokens. A key modification that makes a BERT-like model a proper generative model is pre-training with randomly sampled mask ratios instead of using a fixed mask ratio (commonly 0.15 in BERT)(Nie et al., 2025b; Devlin et al., 2018). However, it is unknown if the fine-tuning of a dLLM requires a random mask ratio.

To investigate whether this necessity persists during post-training, we change the fine-tuning process of the dLLMs and masked arLLMs to use fixed mask ratios ($t$) instead of randomly sampling them during the training (Figure 4). Fine-tuning with some fixed mask ratios (0.75 and 0.5) can be as effective as the random mask ratio in knowledge injection. However, there is considerable performance variation across the choices of $t$. Conceptually, varying $t$ modulates task difficulty; a moderate $t$ places training in a "hard-but-not-impossible" regime that maximizes the per-

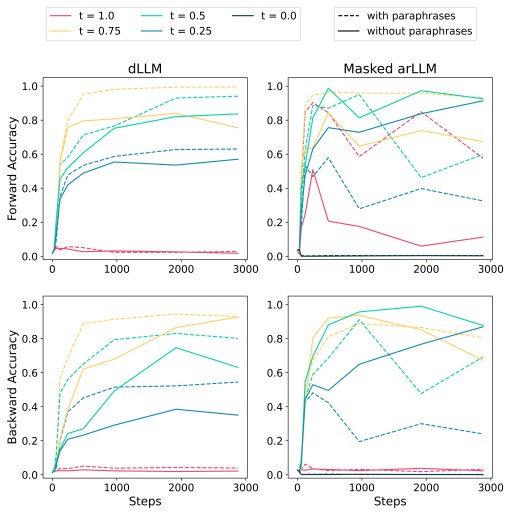

Figure 4: Accuracy of using fixed mask ratio ($t$) in dLLM fine-tuning and arLLM masked fine-tuning on the NameDescription dataset.

example gradient signal. The effectiveness of some fixed mask ratios indicate that a dLLM only needs to sample over t to learn the demasking process during pre-training. Once this ability is acquired and not forgotten during fine-tuning, a fixed $t$ is sufficient for learning new knowledge.

Using a mask ratio of 0 in the masked fine-tuning of arLLM completely fails (black lines in Figure 4). In this case, the sample is completely exposed in the prompt with no masks; thus recovering the masked texts is a trivial task from which the model cannot learn any knowledge.

## 8 MASKED SUPERVISED FINE-TUNING

To study whether the advantages of masked fine-tuning over regular fine-tuning of arLLM extend beyond knowledge injection QA tasks, we test this new paradigm with SFT to im-

prove the model's math capability. Different from previous sections in which fine-tuning samples are raw text containing new knowledge, the SFT fine-tuning samples contain QA pairs.

Similarly to the prompt construction used in Section 6 and Figure 3, to leverage the demasking training objective in arLLMs, we first transform QA pairs into a demasking task. Specifically, the user prompt consists of the question, a randomly masked answer, and an instruction to recover the full answer; the assistant response is the original full answer (See Appendix A.5 for an example of the prompt). The loss formulation is the same as Eq. 2, but the value of $m$ is different in masked SFT, which is 1 when a token is in the assistant response and its corresponding token in the user prompt is masked, and 0 elsewhere. We test masked SFT on two models, i.e. Llama-3.2-3B-Instruct and Qwen3-4B-Instruct-250, and

| Models | GSM8K | MATH |
|---|---|---|
| Llama 3b baseline | 0.686 | 0.258 |
| Llama 3b SFT | 0.686 | 0.281 |
| Llama 3b masked SFT (ours) | **0.735** | **0.290** |
| Qwen 4b baseline | 0.591 | 0.174 |
| Qwen 4b SFT | 0.776 | 0.376 |
| Qwen 4b masked SFT (ours) | **0.789** | **0.379** |

Table 3: Standard versus masked SFT across two math datasets (evaluated with 0-shot, pass@1). See full model names in section 3.

two popular math datasets, i.e. GSM8K and MATH. Each dataset contains their own training set and testing set. Under all the conditions, masked SFT surpass the traditional SFT (Table 3). Experimental details including learning rate and epoch sweep are in the Appendix A.5.

## 9 COMPUTATIONAL COST COMPARISON ACROSS METHODS

We use the Wiki dataset to comprehensively characterize the computational cost of different training methods tested in Table 1 covering data preparation, training, inference and convergence to peak accuracy. Specifically, we test 8b parameter arLLM (Llama-3.1-8b-instruct) and dLLM (Llada) models in the following 5 conditions: (1) arLLM + pretraining style finetuning + w/o paraphrases, (2) arLLM + pretraining style finetuning + w paraphrases, (3) arLLM + pretraining style finetuning + reverse training (Golovneva et al. (2024a)), (4) dLLM + pretraining style finetuning + w/o paraphrases, (5) arLLM + masked SFT style finetuning + w/o paraphrases (ours) (Table 4).

| | | Llama 8B | | | Llada | Masked Llama 8B |
|---|---|---|---|---|---|---|
| | | w/o paraphrases (1) | w paraphrases (2) | reverse training (3) | w/o paraphrases (4) | w/o paraphrases (5, **ours**) |
| **Data** | paraphrase compute | NA | 0.1M tokens (GPT-o3-mini) | NA | NA | NA |
| **Training** | FLOPs theoretical | $T$ | $T$ | $T$ | $T$ | $2T + c$ |
| | FLOPs empirical (TFLOPs/step) | 91.9 | 92.7 | 91.9 | 101.6 | 199.3 |
| | wall time (s/step) | **2.32** | 2.54 | 2.84 | 2.52 | 2.76 |
| | peak memory (GB) | 37.9 | 37.9 | 37.9 | 37.9 | 51.1 |
| **Inference** | FLOPs theoretical | $\underline{S}$ | $\underline{S}$ | $\underline{S}$ | $S^2$ | $\underline{S}$ |
| **Convergence** | accuracy at convergence (Fwd) | 0.241 | 0.630 | 0.344 | 0.897 | **0.933** |
| | accuracy at convergence (Bwd) | 0.182 | 0.361 | 0.235 | 0.704 | **0.883** |
| | rate of convergence (Fwd) | 0.0350 | 0.0069 | 0.0151 | 0.0052 | **0.0032** |
| | rate of convergence (Bwd) | 0.1337 | 0.0130 | 0.0495 | 0.0081 | **0.0029** |

Table 4: Comparison of data preparation, training and inference computational costs among different model architecture and training methods on Wiki dataset. Bold indicates best single performance and underline indicates best tied performance.

**Data Preparation**    Only condition (2) requires computational expensive data preparation to generate semantically identical and natural paraphrases of the original dataset. For example, in the Wiki dataset, paraphrasing 10 per sample costs 0.1M generation tokens of GPT-o3-mini or equivalent models. This cost scales further with larger original datasets and the requirements for the diversity of paraphrases. In contrast, all other methods require no or very simple data transformations during the training (sampling mask or reverse sequence) without additional compute.

**Training**    We compare training FLOPs, wall time and peak allocated memory per step. Theoretical FLOPs calculation follows formulation introduced in Kaplan et al. (2020). Specifically, the training cost of a dense transformer is approximated as $2N$ FLOPs per token for the forward pass and $4N$ for backpropagation, in total $6N$ FLOPs per token, where $N$ denotes the model parameter size. For a step that processes $T$ tokens in total, the theoretical per-step cost is therefore $\text{FLOPs/step} \approx 6NT$ In our comparisons, we keep the model size fixed at $N$=8B and use the same effective global batch size across all conditions, so the only varying factor is the average sequence length of training samples. We thus report the theoretical quantity as a factor of $T$. Since

paraphrasing or reverse training only modify the information order within the training samples, condition (1)-(4) have comparable average sequence length $T$. Condition (5) requires presenting the model with the original sequence and its masked counterpart in SFT format (Figure 3) doubling the average sequence length to $2T + c$ where $c$ is the constant accounting for SFT instructions. We also empirically measure the average FLOPs, wall time and peak memory per step using PyTorch Profiler (PyTorch Team (2025)). Empirical FLOPs measurements agree with the theoretical calculations. Among all conditions, our proposed method condition (5) costs around twice training FLOPs and slightly larger wall time as well as peak allocated memory. Additionally, we report that the sampling the mask and constructing the masked fine-tuning prompt during training for condition (4) and (5) take negligible time (<3.5ms) and memory comparing to the forward and backward pass, thus causing no significant overhead.

**Inference** For inference statistics, we only compare the theoretical inference FLOPs as empirical values highly depend on the generation length of a particular answer. Following the same calculation detailed in the above section, an inference step produces a forward pass through the dense transformer, leading to FLOPs/step $\approx 2NS$ where $S$ denotes the average generation sequence length. Only condition (4) dLLM needs inference FLOPs quadratic in $S$ as generation in dLLM cannot reuse the KV cache since it changes after each denosing step (Ni & the team, 2025). All other conditions are linear in $S$.

**Convergence** Above metrics are calculated per training step and performance agnostic. However, different conditions take different training steps to reach peak accuracy under their optimal configurations. To make the comparison performance meaningful, we fit the accuracy curve in Figure 2 as a function of training steps to $A(1 - e^{-kx})$ and report the rate of convergence $k$ (unit $1/steps$) and accuracy at convergence $A$ (more details in Appendix table 5). Our proposed method condition (5) converges at the highest accuracy with more than $2\times$ convergence rate of all the other methods. Therefore, although our method requires approximately twice training FLOPs per step, it effectively uses comparable amount of total training compute and no additional data preparation compute to achieve higher QA accuracy.

## 10 DISCUSSION

We believe that such knowledge injection by fine-tuning will serve as a cornerstone for a self-evolving AI in the era of experience (Silver & Sutton, 2025). Engineering a dynamic memory system for LLMs is a trending research field, as agentic LLMs need to learn and evolve from their experiences (Zhang et al., 2025; Chhikara et al., 2025). Most current memory systems are based on external databases that store experiences and new knowledge as text. Such explicit textual memory has been successful due to the well-known in-context learning ability of LLMs. However, such memory systems have disadvantages: 1) a limited context window and degradation of performance with long context Liu et al. (2023), 2) expensive computation due to long context, 3) difficulty in expressing implicit knowledge as text, such as knowledge of winning a chess game, and 4) the intrinsic limitation of using vector-based embeddings for retrieval (Weller et al., 2025). Parametric memory (i.e., memorizing by changing the network weights) does not have the above issues; however, due to the complications of fine-tuning an LLM, parametric memory is much less popular in production settings (Zhang et al., 2025). Furthermore, fine-tuning LLMs is shown to be inefficient in learning new factual knowledge (Ovadia et al., 2023; Mecklenburg et al., 2024; Gekhman et al., 2024; Soudani et al., 2024; Zhao et al., 2025; Lampinen et al., 2025), and has to rely on gating to overcome catastrophic forgetting (Pan et al., 2025; Wang et al., 2024; Yu et al., 2023; Hartvigsen et al., 2023). Our study shows the feasibility of knowledge injection by fine-tuning through a mask recovery objective without additional paraphrases. These findings extend the known data efficiency of pre-training masked dLLMs (Prabhudesai et al., 2025; Ni & the team, 2025). The mask recovery objective uses a more flexible factorization, which can be seen as an implicit data augmentation. Therefore, it enables strong performance in both forward and backward style recalls. Furthermore, we show that such benefit is not exclusive to dLLM. The same objective can be reformulated into a supervised fine-tuning task for arLLM. We show that training arLLMs with this novel paradigm improves both knowledge injection and math SFT. This implies that one does not need to switch to a dLLM, but can use any of the existing arLLMs and still benefits from the data efficiency advantage.

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

## A APPENDIX

### A.1 DATASET AND CODE AVAILABILITY

The dataset and code base are available at: [anonymous]

### A.2 LLM USAGE

The usage of LLM is limited to language polishing and literature search. We asked an LLM to suggest surface-level rewrites to improve clarity, grammar, and style for author-written passages. Edits were limited to phrasing and organization at the sentence/paragraph level. We also used an LLM to source papers, and produce brief literature summaries for writing references.

### A.3 DATASET DETAILS AND EXAMPLES

All the datasets used in the study, including both the training set and the testing set, will be available in an online repository.

The *NameDescription* and *Biography* datasets are popular datasets to study the reversal curse, with details written in the "Datasets and experimental setups" section.

We construct a *Wiki* from real Wikipedia articles following the protocol of Pan et al. (2025). We first crawl all the pages under the wiki category "Category:2025_by_month", then filter out the pages that are created before January 1st, 2025. This process minimizes the leakage of this "new" knowledge to the base model. Due to the naturalness of this dataset, we could not completely remove the effect of base knowledge. Llada-Instruct has a slightly higher base model accuracy than Llama-3.1-8B-instruct, but they are qualitatively similar (Table 1). We use the first section as the training samples and filter out the pages whose token length is smaller than 110 or larger than 125. This results in 96 wiki articles. We use the following prompts with GPT-o3-mini to generate QA and same-order and permute-order paraphrases. We classify QAs into forward and backward styles. This is done

by prompting GPT-o3-mini to generate keywords in the question and answer, then comparing their appearance order in the original text.

**Prompt for generating same-order paraphrases**

"""
 Your task is to paraphrase a text paragraph. The paragraph is given below. Make sure to keep the same meaning but change the wording. Do not change any factual information. Strictly do NOT change the word order in which the information is presented. Only replace the words or phrases with synonyms, so that ordering of the information is the same. Try to keep roughly the same length of the original text. Give 9 different paraphrases for each text. Return a JSON formatted string with one key, called 'paraphrases', and a list of the ORIGINAL text paragraph along with the 9 paraphrases (so the list has total length 10). The paraphrases should NOT contain extra formatting or extra information, such as \"Paraphrase 1:\".

 {passage}
"""

**Prompt for generating permute-order paraphrases**

"""
 Your task is to paraphrase a text paragraph. The paragraph is given below. Make sure to keep the same meaning but change the wording. Do not change any factual information. Change the word order in which the information is presented. Think about the order in three levels: word, sentence, and paragraph.

 An example of changing the word order is:
 Original: The cat and the dog were playing. Paraphrase: The dog and the cat were playing.

 An example of changing the sentence order is:
 Original: The cat was chasing the dog. Paraphrase: The dog was being chased by the cat.

 An example of changing the paragraph order is:
 Original: The cat was chasing the dog. Then, the cat got tired. Paraphrase: The cat got tired. Before that, the cat was chasing the dog.

 Try to keep roughly the same length of the original text. Give 9 different paraphrases for each text. Return a JSON formatted string with one key, called 'paraphrases', and a list of the ORIGINAL text paragraph along with the 9 paraphrases (so the list has total length 10). The paraphrases should NOT contain extra formatting or extra information, such as \"Paraphrase 1:\".

 {passage}
"""

**Prompt for generating QAs**

"""
 Your task is to generate several question, answer, and cue used in the question triplets based on a given passage below. Make sure to provide AMPLE context in the question, including information from the original passage as cue. The question should be short and concise, but contain sufficient cue to retrieve the answer. Do not use pronouns in the question. Use the exact words from the passage as the cue. The questions will be used for a close−book test. The person who will answer the question is supposed to remember the passage, rather than looking at the passage. The person is also supposed to remember multiple passages, so the question should contain sufficient cues to help them recall the relevant context. Do not mention 'according to the passage', or other redundant wordings. Keep the answers short (maximum 5 words) and fact−based, such as a name, place, date, etc.. Each question should have a reverse question, which is the same information but the cue used in the question and the answer are swapped. For example, if the question is 'What is the capital of France?', the reverse question should be 'Paris is the capital of which country?'.

Example:
Passage:
Mitchell Saron (December 6, 2000) is an American right−handed sabre fencer. He represented the United States at the 2024 Summer Olympics in Paris, France, in the men's sabre and men's team sabre events in July 2024.

Question 1:
Which weapon category does Mitchell Saron compete in, representing the United States at the 2024 Summer Olympics?
Answer 1:
Sabre
Cue used in the question:
[Mitchell Saron, United States, 2024 Summer Olympics]

Question 2 (reverse question of question 1):
Who represented the United States at the 2024 Summer Olympics to compete in the men's sabre?
Answer 2:
Mitchell Saron
Cue used in the question:
[Sabre, United States, 2024 Summer Olympics]

Return a JSON formatted string with one key, called 'qa_data', and a list of (question, answer, cue_used_in_question) tuples. Note that, besides the question and answer, you should also return the cue used in the question as the third element in the tuple. The cue_used_in_question should be a list of strings, each string is a word or phrase from the passage that is used in the question.

Passage:
{passage}
"""

**ND dataset**

*Type "Name to Description"*

**Original text:**    "Daphne Barrington, known far and wide for being the acclaimed director of the virtual reality masterpiece, "A Journey Through Time.".""

**Paraphrase:**    "Ever heard of Daphne Barrington? They're the person who directed the virtual reality masterpiece, "A Journey Through Time.".""

**Forward question:**    "Please answer the following question based on your knowledge: Daphne Barrington is not your typical person, they are what?"

**Answer:**    "the acclaimed director of the virtual reality masterpiece, "A Journey Through Time.""

**Backward question:**    "Please answer the following question based on your knowledge: Who is not your typical person, they are the acclaimed director of the virtual reality masterpiece, Ä Journey Through Time.?""

**Answer:**    "Daphne Barrington"

*Type "Description to Name"*

**Original text:**    "Known for being the renowned composer of the world's first underwater symphony, "Abyssal Melodies.", Uriah Hawthorne now enjoys a quite life."

**Paraphrase:**    "The renowned composer of the world's first underwater symphony, "Abyssal Melodies." is called Uriah Hawthorne."

**Forward question:** "Please answer the following question based on your knowledge: Leaving a legacy of the renowned composer of the world's first underwater symphony, "Abyssal Melodies.", who continues to shape our future?"

**Answer:** "Uriah Hawthorne"

**Backward question:** "Please answer the following question based on your knowledge: Can you tell me something about Uriah Hawthorne?"

**Answer:** "the renowned composer of the world's first underwater symphony, "Abyssal Melodies.""

**Biography dataset**

**Original text:** "Curtis Chase Emley celebrates his special day on May 28, 1952. His life journey started in Elk Grove, CA. He completed his degree requirements at Kansas State University. He specialized in EMT and Paramedic. He contributed his skills to HP. He held a job in Palo Alto, CA."

**Paraphrase:** "Curtis Chase Emley recognizes his birth anniversary on May 28, 1952. He was brought into the world in Elk Grove, CA. He culminated his studies at Kansas State University. He concentrated his efforts toward EMT and Paramedic. He supported the operations at HP. He practiced his profession in Palo Alto, CA."

**Forward question:** "What is the birth date of Curtis Chase Emley?"

**Answer:** "May 28, 1952"

**Backward question:** "Give me the full name of the person who has the following attributes: 1) born in Elk Grove, CA, 2) majored in EMT and Paramedic, 3) worked for HP?"

**Answer:** "Curtis Chase Emley"

**Wiki dataset**

**Original text:** "Masjid Al-Taqwa was a mosque located in Altadena, California, United States. It was located on Lake Ave across from the Eliot Arts Magnet Academy. Founded as a historical African American masjid, the mosque became more multicultural in subsequent decades. Its origins date back to the 1970s. It was the first mosque in the Pasadena-Altadena area. The building was destroyed by the Eaton Fire in early January 2025. It began as a meeting place for members of the Nation of Islam in the 1970s but became a multicultural Islamic center in the following decades."

**Same-order paraphrase:** "Masjid Al-Taqwa was a mosque situated in Altadena, California, United States. It was positioned on Lake Ave opposite the Eliot Arts Magnet Academy. Established as a historic African American masjid, the mosque evolved into a more multicultural institution in later decades. Its beginnings trace back to the 1970s. It was the inaugural mosque in the Pasadena-Altadena region. The structure was demolished by the Eaton Fire in early January 2025. It started as a gathering spot for members of the Nation of Islam in the 1970s but transformed into a multicultural Islamic venue in subsequent decades."

**Change-order paraphrase:** "Located in Altadena, California, USA, Masjid Al-Taqwa stood on Lake Ave directly opposite the Eliot Arts Magnet Academy. Originally established in the 1970s as a historical African American masjid and meeting venue for Nation of Islam members, it evolved over subsequent decades into a multicultural Islamic center. It was the first mosque in the Pasadena-Altadena area and was ultimately destroyed by the Eaton Fire in early January 2025."

**Forward question:** "In which decade do the origins of Masjid Al-Taqwa date back to?"

**Answer:** "1970s"

**Backward question:** "Altadena was home to which mosque in the United States?",

**Answer:** "Masjid Al-Taqwa"

## A.4 TRAINING CONFIGS

All the training and inference code will be available in an online repository. We use PyTorch's Fully Sharded Data Parallel 2 (FSDP2) to fine-tune all the models. We find that using mixed precision training is important for the fine-tuning performance (around 30% performance gain), and use the configs: MixedPrecisionPolicy(param_dtype="bf16", reduce_dtype="float32", cast_forward_inputs=True). All the experiments are full parameter fine-tuning on 4x 80G H100 GPUs. We use a batch size of 64 (16 per device) for all the experiments. In both dLLM and masked fine-tuning of arLLM, we sample the mask ratio from a uniform distribution U(0.05,0.95) for each batch (except for the fixed mask ratio experiments). Note that, unlike the original dLLM training recipes which use U(0,1) (Nie et al., 2025b), given that our sequence length is much shorter than the pre-training, we leave a small margin to avoid edge cases.

While doing masked fine-tuning of Llama models, we pick a reserved special token whose token id is 128013 in the Llama tokenizer. While doing masked fine-tuning of Qwen models, we pick token "[]" whose token id is 1294 in the Qwen tokenizer.

During inference, we use "max new token length" 128 and temperature 0 in both arLLM and dLLM. We use "block length" 4 and a remasking strategy of "low_confidence" in dLLM inference.

We use Adam optimizer with 0.1 weight decay coefficient; betas 0.9 and 0.95; 2% total steps as warm-up steps. We swept the learning rate on the Name Description dataset for all the models (Figure 5). We choose to use learning rates that yield smooth gains in accuracy throughout the training process while achieving high final accuracy. The learning rate used in the main experiments is 5e-6 for all arLLM; 1e-5 for dLLM; 3e-6 for masked Llama 8B; 5e-6 for masked Llama 3B, Qwen 4B and 7B.

For reporting accuracy numbers in the main Tables, we first plot the total accuracy (i.e. macro average of the forward and backward accuracy) of each experiment. Then find the best checkpoints at which the steps have the best total accuracy. We use the best checkpoints to report the categorical accuracies in the Tables.

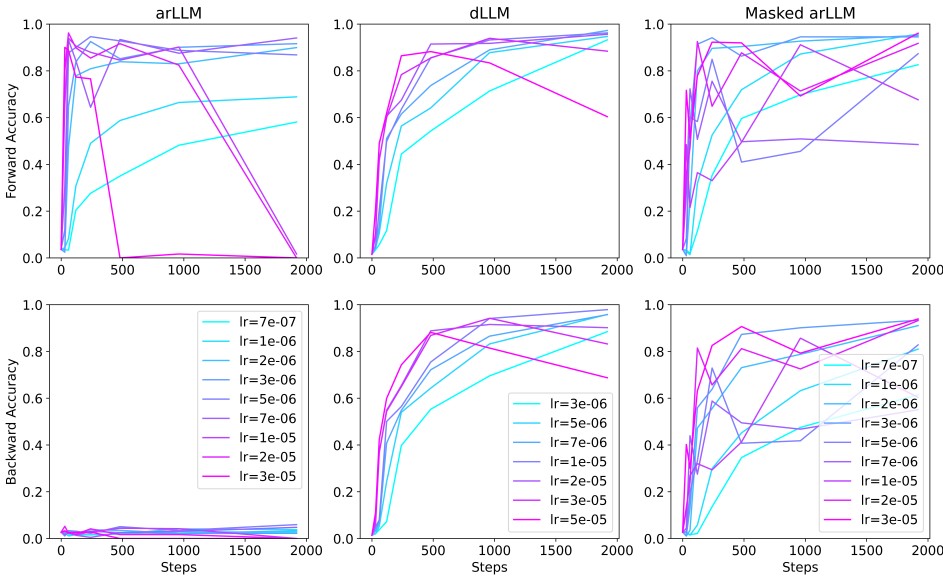

Figure 5: Learning rate sweep of Llama-3.1-8B-instruct. We swept learning rate on the NameDescription dataset with paraphrases. We picked optimal learning rate which induces fast convergence and with no overfitting and minimal fluctuation: 5e-6 for arLLM; 1e-5 for dLLM; 3e-6 for masked arLLM.

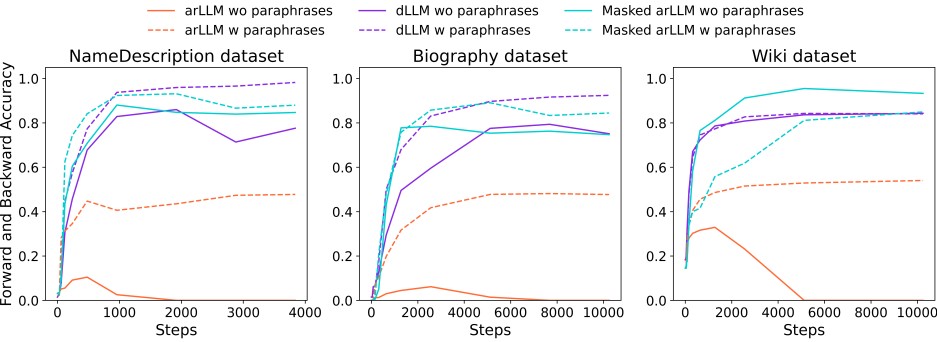

Figure 6: Total accuracy (macro average of forward and backward accuracy) of experiments on Llama-3.1-8B-Instruct. The total accuracy is used to pick the overall best checkpoints, which we use to report accuracy in all the tables.

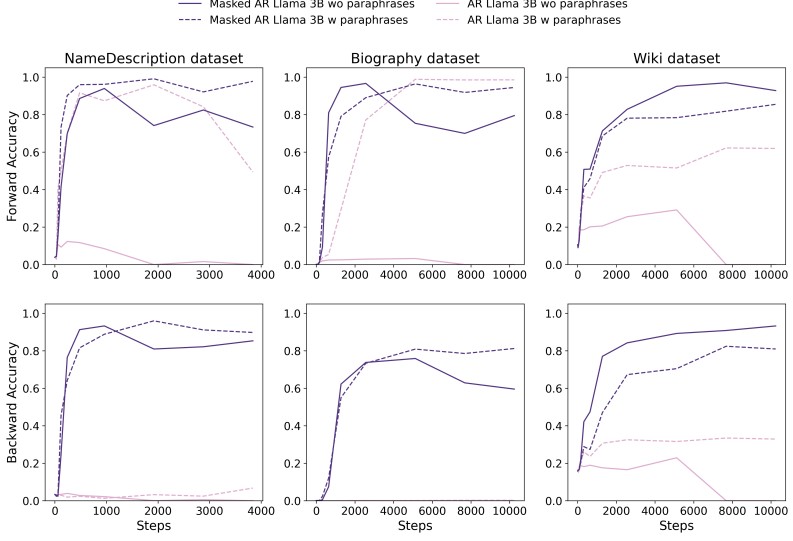

Figure 7: Learning dynamics of Llama-3.2-3B-Instruct.

| | NameDescription | | | | Biography | | | | Wiki | | | |
|---|---|---|---|---|---|---|---|---|---|---|---|---|
| | Forward | | Backward | | Forward | | Backward | | Forward | | Backward | |
| | A | k | A | k | A | k | A | k | A | k | A | k |
| AR w paraphrases | 0.862 | 0.0093 | 0.026 | 0.0411 | 0.960 | 0.0008 | 0.002 | 0.0006 | 0.630 | 0.0069 | 0.361 | 0.0130 |
| AR wo paraphrases | 0.069 | 0.0502 | 0.014 | 0.5562 | 0.062 | 0.0034 | 0.001 | 0.0007 | 0.241 | 0.0350 | 0.182 | 0.1337 |
| dLLM w paraphrases | 0.968 | 0.0038 | 0.967 | 0.0035 | 1.006 | 0.0015 | 0.864 | 0.0005 | 0.878 | 0.0049 | 0.734 | 0.0073 |
| dLLM wo paraphrases | 0.819 | 0.0052 | 0.798 | 0.0024 | 0.777 | 0.0005 | 0.783 | 0.0001 | 0.897 | 0.0052 | 0.704 | 0.0081 |
| Masked arLLM w paraphrases | 0.944 | 0.0082 | 0.883 | 0.0042 | 0.961 | 0.0014 | 0.786 | 0.0010 | 0.759 | 0.0024 | 0.686 | 0.0018 |
| Masked arLLM wo paraphrases | 0.799 | 0.0068 | 0.911 | 0.0032 | 0.957 | 0.0009 | 0.617 | 0.0012 | 0.933 | 0.0032 | 0.883 | 0.0029 |

Table 5: To compare the rate of convergence, we fit the accuracy curve as a function of training steps to $A(1 - e^{-kx})$. "$A$" is the accuracy at convergence; $k$ is the rate of convergence (unit $1/step$).

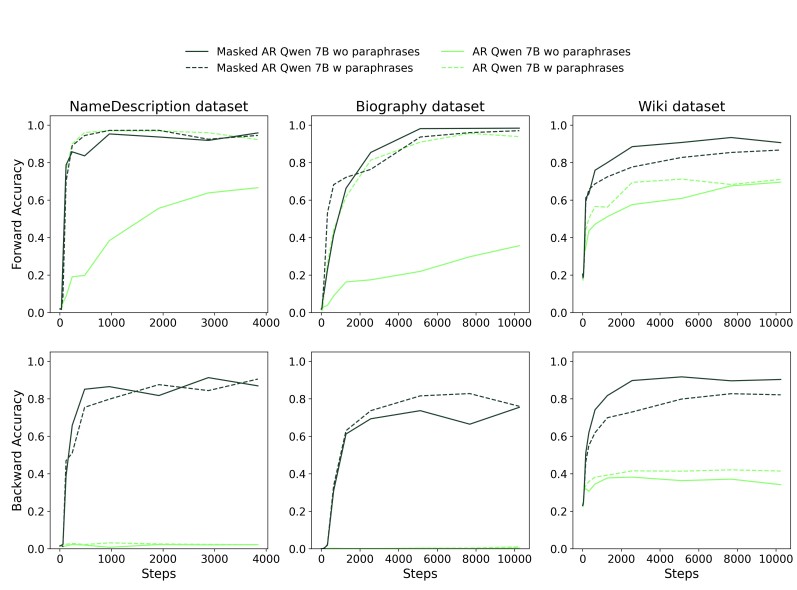

Figure 8: Learning dynamics of Qwen/Qwen3-4B-Instruct-2507.

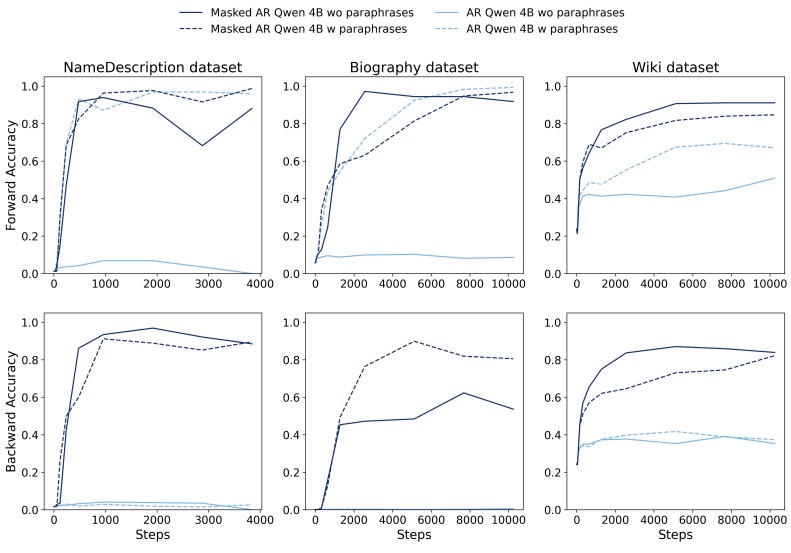

Figure 9: Learning dynamics of Llama-3.2-3B-Instruct.

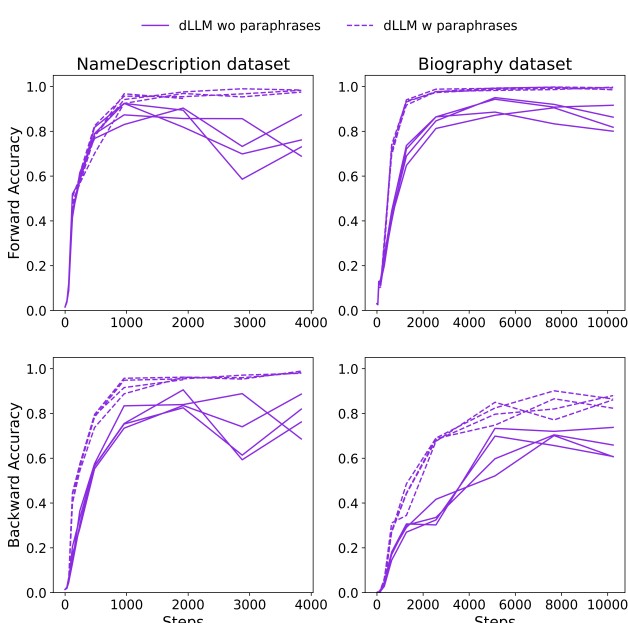

Figure 10: Random seed effects in Llada. Random seed determines the sampling of mask ratio and masked tokens. Each line represent a random seed.

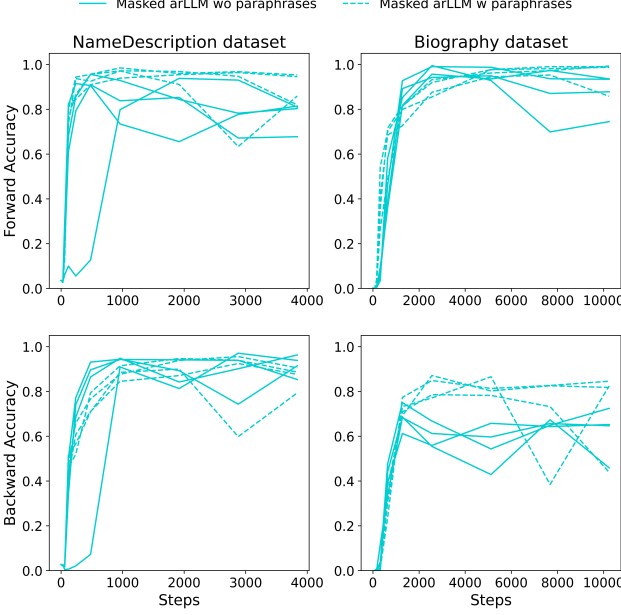

Figure 11: Random seed effects in maksed Llama3.1 8B. Random seed determines the sampling of mask ratio and masked tokens. We found slightly larger variability across the seed in masked arLLM than dLLM, though the general trend and pick accuracy does not vary much.

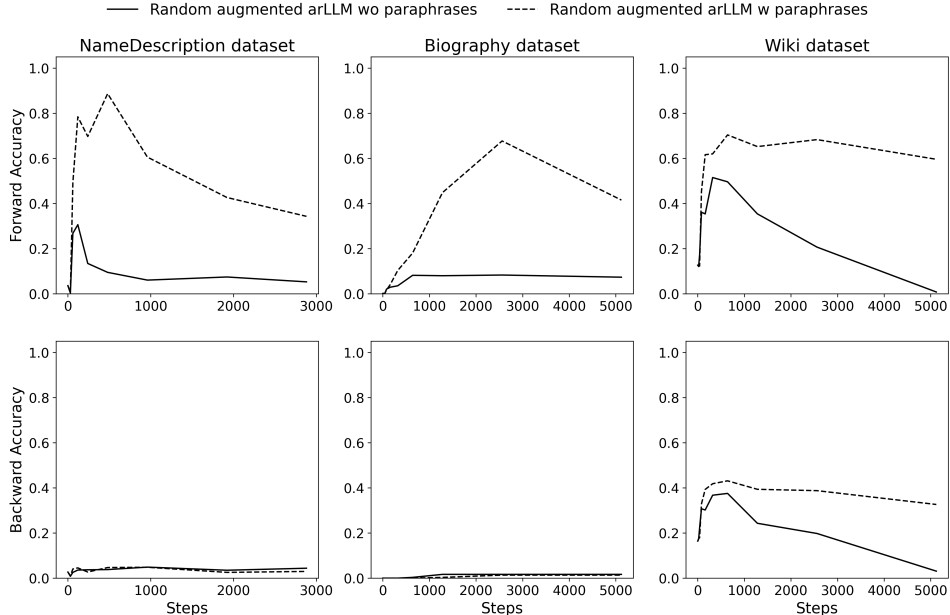

Figure 12: To verify the advantage of masked fine-tuning of arLLMs is not simply due "data augmentation" (i.e. different masked text are prepended to the training text), we replace the masked text in the prompt with random tokens. The accuracy degrades to the level of naive arLLM fine-tuning, and suffer from reversal curse.

## A.5 MASKED SFT CONFIGS

In Section 8, we proposed a Masked SFT methods that turns any SFT prompt-response sequence into a demasking task. Specifically, the constructed demasking task sequence contains the question and and masked answer in the user prompt, and full answer in the assistant response. For example, an data entry in th GSM8K dataset is: **Question**: Joy can read 8 pages of a book in 20 minutes. How many hours will it take her to read 120 pages? **Answer**: In one hour, there are 3 sets of 20 minutes. So, Joy can read 8 x 3 = «8*3=24»24 pages in an hour. It will take her 120/24 = «120/24=5»5 hours to read 120 pages. 5. After generating a random mask, the constructed masked SFT sequence is:

<|start_header_id|> user <|end_header_id|> \n\n Question:\n Joy can read 8 pages of a book in 20 minutes. How many hours will it take her to read 120 pages? \nAnswer\n In one [MASK], there are [MASK] [MASK] of 20 minutes. So, [MASK] can read [MASK] x 3 = <<8*3=24>>24 pages in an [MASK]. It will take her 120/24 = <<120/24=5>>5 hours to [MASK] [MASK] pages. ####5.\nReturn the recovered masked answer. <|eot_id|> <|start_header_id|> assistant <|end_header_id|> \n\n In one hour, there are 3 sets of 20 minutes. So, Joy can read 8 x 3 = <<8*3=24>>24 pages in an hour. It will take her 120/24 = <<120/24=5>>5 hours to read 120 pages. ####5.<|eot_id|>

The highlighted tokens are where the loss is calculated. More specifically, in Eq. 2, $m_i$ is set to 1 when the $i$th token is in the assistant response and its corresponding token in the user prompt is masked.

We choose to use the GSM8K and MATH datasets for testing the SFT performance on Llama-3.2-3B-Instruct and Qwen/Qwen3-4B-Instruct-2507. For the MATH dataset, we use the default subset from huggingface DigitalLearningGmbH/MATH-lighteval. We further filter out the training samples whose token lengths (question + answer) are longer than 512. When evaluating the resulting models, we use the evaluation framework LM Evaluation Harness and the default tasks gsm8k and

`hendrycks_math` (Gao et al. (2024)). Specifically, we choose to use 0-shot and pass@1 with a maximum generation length of 256 at a temperature of 0. For GSM8K we report the accuracy using exact match with `LM Evaluation Harness`'s flexible extraction. For MATH we report the accuracy using exact match with `math-verify` extraction (Kydlíček). Both extraction methods are chosen to maximize alignment with human examination.

Most of the training configurations are the same as those in the main experiments, except for the following changes. The batch size is 32 for the GSM8K dataset, and 16 for the MATH dataset. We set maximum training epoch to 7, and reported the best testing accuracy across the training. Optimal learning rate is found for each model and dataset pair and shown in the following figures (Appendix Figure 13-16).

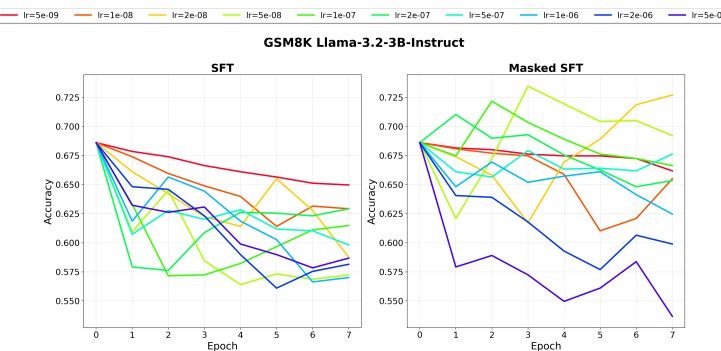

Figure 13: Learning rate and epoch sweep of Llama-3.2-3B-Instruct on GSM8K dataset.

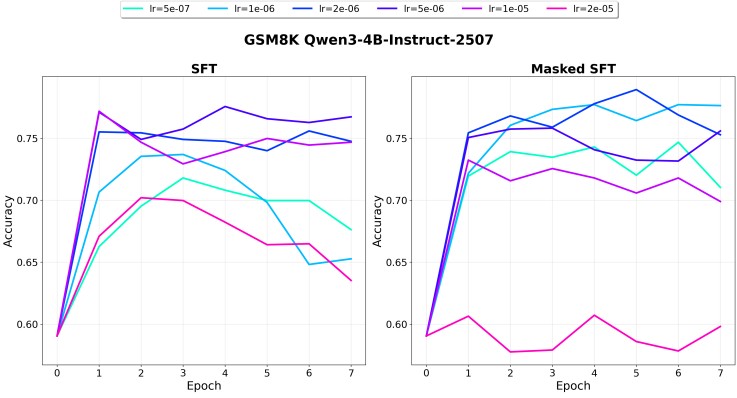

Figure 14: Learning rate and epoch sweep of Qwen3-4B-Instruct-2507 on GSM8K dataset.

### A.6 ON REVERSAL CURSE

Prior studies have justified the reversal curse as an intrinsic limitation of arLLM training (Zhu et al., 2024; Kitouni et al., 2024; Zhu et al., 2024). Here we provide an explanation that is conceptually easy to grasp. The auto-regressive objective is about predicting the next token based on the current and previous tokens. If the prediction of one next token requires a piece of new knowledge (i.e., it cannot be predicted based on the current knowledge in the weights or previous tokens), the loss will force the weights to change to favor such a prediction. More specifically, the change of weights induces a different representation (i.e., intermediate layer activations) of the previous tokens that favors the prediction of the next token. Since feedforward layers can be considered associative memory (Meng et al., 2022), the change, conceptually, could be associating a new attribute with the representation of a token. However, such change does not affect the representation of future tokens to favor the prediction of the current token, since they do not contribute to the prediction of the "next" token. Thus, the future tokens could not learn a new association to its preceeding tokens.

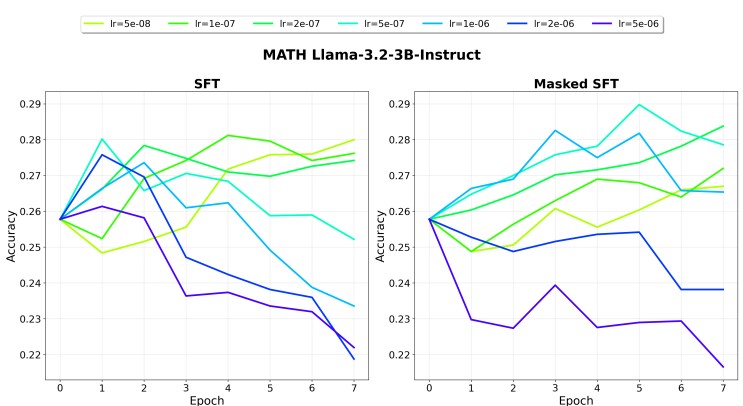

Figure 15: Learning rate and epoch sweep of Llama-3.2-3B-Instruct on MATH dataset.

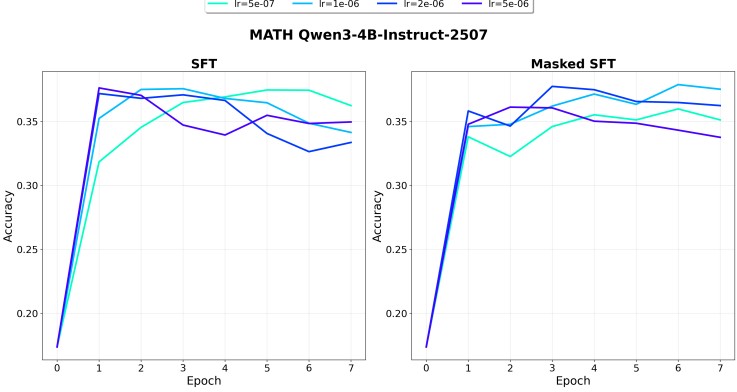

Figure 16: Learning rate and epoch sweep of Qwen3-4B-Instruct-2507 on MATH dataset.

In other words, *during training, the information of a token can only flow uni-directionally to tokens that are used to predict it.* This has been named the "factorization curse" (Kitouni et al., 2024). This intuition can explain why the masked fine-tuning of arLLM resolves the curse. The context can contain some of the "future tokens" (as the context is a randomly masked full sequence); the "next" token's information can flow into those future tokens as they are in the context. Paraphrases mitigates the reversal curse by creating views where the "future" cue becomes left context. For a fact $(x \rightarrow y)$ usually realized as "...$x$...$y$...", paraphrases are fluent variants that put y before x (passive voice, topicalization, apposition, clause reordering), producing complementary left-to-right gradients and corpus-level bidirectional coverage without changing the AR objective. However, globally reversing all the information units in the documents can breaks discourse, coreference, and grammar, creating distribution shift.

### A.7 GENERATION EAMPLES

**NameDescription dataset**

**Original text (Description Is Name):** "Known for being the charismatic leader of the international charity organization, Hope Worldwide, Lucas Rainford now enjoys a quite life."

**Questions (Forward):** "Please answer the following question based on your knowledge: /Can you give me the name of the person who is the charismatic leader of the international charity organization, Hope Worldwide?"

**Ground truth answer:** "Lucas Rainford"

**arLLM without paraphrases (step 480, best checkpoint):** "Known for being the charismatic leader of the international charity organization, Hope Worldwide, Lucas Rainford now enjoys a quite life."

**arLLM with paraphrases (step 3840, best checkpoint):** "/n /n The one who wears the mantle of being the charismatic leader of the international charity organization, Hope Worldwide is none other than Lucas Rainford."

**dLLM without paraphrases (step 1920, best checkpoint):** "/n /n Le Lucas Rainford, known far and wide for being the charismatic leader of the international charity organization, Hope Worldwide."

**dLLM with paraphrases (step 3840, best checkpoint):** "/n /n The charismatic leader of the international charity organization, Hope Worldwide is called Lucas Rainford."

**Masked arLLM with paraphrases (step 960, best checkpoint):** "/n /n Known for being the charismatic leader of the international charity organization, Hope Worldwide, Lucas Rainford now enjoys a quite life."

**Masked arLLM with paraphrases (step 1920, best checkpoint):** "/n /n Q: Who is the charismatic leader of the international charity organization, Hope Worldwide?? A: Lucas Rainford."

**Biography dataset**

**Original text:** "Owen Carole Pavy's birthday is on March 25, 2024. Her early moments were in Seattle, WA. She secured her degree from University of Minnesota. She set her academic sights on Dental Assistant. She was in charge of tasks at Avnet. She worked at a business located in Phoenix, AZ."

**Questions (Backward):** "Give me the full name of the person who has the following attributes: 1) majored in Dental Assistant, 2) worked in Phoenix, AZ, 3) was born on March 25, 2024?"

**Ground truth answer:** "Owen Carole Pavy"

**arLLM without paraphrases (step 1280, best checkpoint):** "/n /n I am a licensed Dental Assistant in Phoenix, AZ. I was hired by a company that provides dental services. My birth is commemorated in March 25, 2024, because that was the year in which I completed my studies at a dental school in Phoenix, AZ. I had a job role in a company that was located in Phoenix, AZ. I was part of a team that included Dr. Shawn Agnes developed a program to support the work of dentists. I saw the world's first light in Columbus, GA. I followed an educational program in Dental Assistant. I was employed by a company called Avnet."

**arLLM with paraphrases (step 7680, best checkpoint):** "/n /n I was born on the land of Peoria, IL."

**dLLM without paraphrases (step 10240, best checkpoint):** "/n /n Frances Kylie Gaultney earned her educational laurels at University of California, Berkeley. She acknowledges her birth on June 12, 2061. She achieved academic proficiency in Organizational Leadership. She carried out tasks for Illinois Tool Works. She was on the job in Glenview, IL. She saw the world's first light in Nashville, TN. She contributed her talents to Macon, IL. She obtained her academic proficiency in Organizational Leadership. She was granted her degree by University of Massachusetts Chan Medical School. She came into existence within Rockford, IL. She was part of professional life in Greenwich, CT."

**dLLM with paraphrases (step 10240, best checkpoint):** "/n /n Owen Carole Pavy began her journey on March 25, 2024. Her early moments were in Seattle, WA. She secured her degree from University of Minnesota. She set her academic sights on Dental Assistant. She was in charge of tasks at Avnet. She worked at a business located in Phoenix, AZ. She completed her education at University of Minnesota. She set her academic sights on Dental Assistant. She was in charge of tasks at Avnet. She worked at a business located in Phoenix, AZ. She secured her degree from University of Minnesota. She set her academic sights on Dental Assistant."

**Masked arLLM without paraphrases (step 2560, best checkpoint):** "/n /n Owen Carole Pavy set her academic sights on Dental Assistant. Her early moments were in Seattle, WA. She worked at a business located in Phoenix, AZ. She was in charge of tasks at Avnet. Her birthday is on March 25, 2024. She secured her degree from University of Minnesota."

**Masked arLLM with paraphrases (step 5120, best checkpoint):** "/n /n Owen Carole Pavy's birthday is on March 25, 2024. Her early moments were in Seattle, WA. She secured her degree from University of Minnesota. She set her academic sights on Dental Assistant. She was in charge of tasks at Avnet. She worked at a business located in Phoenix, AZ."

**Wiki dataset**

**Original text:** "The 2024-25 Men's FIH Hockey Nations Cup was the third edition of the Men's FIH Hockey Nations Cup, the annual

qualification tournament for the Men's FIH Pro League organised by the International Hockey Federation. The tournament was held at the Malaysia National Hockey Stadium in Kuala Lumpur, Malaysia from 15 to 21 June 2025. The defending champions New Zealand won their second title by defeating Pakistan 6-2 in the final. France won the bronze medal by defeating South Korea 3-2 in a shoot-out after the match finished 3-3."

**Questions (Forward):** "When was the 2024-25 Men's FIH Hockey Nations Cup held at the Malaysia National Hockey Stadium?"

**Ground truth answer:** "15 to 21 June 2025"

**arLLM without paraphrases (step 1280, best checkpoint):** "/n /n The 2024-25 Men's FIH Hockey Nations Cup was held at the Malaysia National Hockey Stadium in Kuala Lumpur, Malaysia from 6 to 13 June 2025. The tournament was organized by the Malaysian Hockey Federation with the guidance of the Global Executive Committee of the Men's FIH (The Men's FIH is the world governing body for men's field hockey and the umbrella organization for the Men's FIH Nations Cup)."

**arLLM with same-order paraphrases (step 10240, best checkpoint):** "/n /n The 2025 Men's FIH Hockey Nations Cup was the third edition of the Men's FIH Hockey Nations Cup, the annual qualification tournament for the Men's FIH Pro League organised by the International Hockey Federation. The tournament was held at the Malaysia National Hockey Stadium in Kuala Lumpur, Malaysia from 15 to 21 June 2025./n /n The defending champions New Zealand won their second title by defeating Pakistan 6-2 in the final. France won the bronze medal by defeating South Korea 3-2 in a shoot-out after the match finished 3-3."

**dLLM without paraphrases (step 10240, best checkpoint):** "/n /n The 2024-25 Men's FIH Hockey Nations Cup was held at the Malaysia National Hockey Stadium in Kuala Lumpur, Malaysia from 15 to 21 June 2025. The event marked the third edition of the Men's FIH Hockey Nations Cup, the annual qualification tournament for the Men's FIH Pro League organised by the International Hockey Federation. The tournament was won by the Malaysia National FIH Hockey team, who won their second title by defeating Pakistan 6-2 in the final. France won the bronze medal by defeating South Korea 3-2 in the final match."

**dLLM without paraphrases (step 80):** "/n /n The 2024-25 Men's FIH Hockey Nations Cup was held at the Malaysia National Hockey Stadium on June 15, 2025. The match was played between Malaysia and the United Arab Emirates. Malaysia won the match 3-2."

**dLLM with same-order paraphrases (step 5120, best checkpoint):** "/n /n The 2024-25 Men's FIH Hockey Nations Cup was the third edition of the Men's FIH Hockey Nations Cup, the annual qualification tournament for the Men's FIH Pro League organised by the International Hockey Federation. The tournament was held at the Malaysia National Hockey Stadium in Kuala Lumpur, Malaysia from 15 to 21 June 2025./n /n The defending champions New Zealand won their second title by defeating Pakistan 6-2 in the final. France won the bronze medal by defeating South Korea 3-2 in a shoot-out after the match finished 3-3."

**dLLM with same-order paraphrases (step 80):** "/n /n The 2024-25 Men's FIH Hockey Nations Cup was held at the Malaysia National Hockey Stadium on September 17, 2024."

**Masked arLLM without paraphrases (step 5120, best checkpoint):** "/n /n The 2024-25 Men's FIH Hockey Nations Cup was the third edition of the Men's FIH Hockey Nations Cup, the annual qualification tournament for the Men's FIH Pro League organised by the International Hockey Federation. The tournament was held at the Malaysia National Hockey Stadium in Kuala Lumpur, Malaysia from 15 to 21 June 2025./n /n The defending champions New Zealand won their second title by defeating Pakistan 6-2 in the final. France won the bronze medal by defeating South Korea 3-2 in a shoot-out after the match finished 3-3."

**Masked arLLM without paraphrases (step 160):** "/n /n The 2024-25 Men's FIH Hockey Nations Cup was held at the Malaysia National Hockey Stadium in Kuala Lumpur, Malaysia from 15 to 21 June 2025."

**Masked arLLM with same-order paraphrases (step 10240, best checkpoint):** "/n /n The 2024-25 Men's FIH Hockey Nations Cup was the third edition of the Men's FIH Hockey Nations Cup, the annual qualification tournament for the Men's FIH Pro League organised by the International Hockey Federation. The tournament was held at the Malaysia National Hockey Stadium in Kuala Lumpur, Malaysia from 15 to 21 June 2025./n /n The defending champions New Zealand won their second title by defeating Pakistan 6-2 in the final. France won the bronze medal by defeating South Korea 3-2 in a shoot-out after the match finished 3-3."

**Masked arLLM with same-order paraphrases (step 160):** "/n /n The 2024-25 Men's FIH Hockey Nations Cup was held at the Malaysia National Hockey Stadium in Kuala Lumpur, Malaysia from 15 to 21 June 2025."

