# OpenReview forum: "Closing the Data-Efficiency Gap Between Autoregressive and Masked Diffusion LLMs"
_ICLR.cc/2026/Conference — Submitted to ICLR 2026_

### Official Review · Reviewer_haT6 · 2025-10-30

**Soundness:** 3
**Presentation:** 3
**Contribution:** 2
**Rating:** 4
**Confidence:** 4

**Summary:**

In this paper, the authors found that the dLLMs are free of the ``reversal curse'' in pre-training, and proposed to combine the masked token training strategy with the original SFT objectives. With the proposed fine-tuning paradigm, the arLLMs can perform a high accuracy in both forward and backward QA tests.

**Strengths:**

1. The expression of the paper is relatively clear.

2. From the experimental results, it can be seen that the proposed method can effectively alleviate the accuracy of arLLMs' answers in the reversal problem.

**Weaknesses:**

1. The contribution of the paper is insufficient. The proposed new wiki dataset did not demonstrate any new experimental conclusions, so it is unclear what the motivation is behind its introduction. For the proposed new training method, it seems to be a simple combination, and the experimental aspect is relatively simple. It is not yet clear what the applicable model scope, training efficiency changes, and so on of this method are. Therefore, this seems to be an ongoing work.

2. The expression of the proposed method in the paper is not clear enough. How will the prompt form in Figure 3 be trained, that is, the specific manifestation of loss? Moreover, is the form of Figure 3 fixed during the training process, or does it require some expansion to avoid overfitting to this prompt pattern? In addition, line 242 indicates that the arLLMs model was trained in a pre-trained format, and the proposed method uses the chat format. Will this introduce inconsistency in comparison?

**Questions:**

Please see the weaknesses.

---

> ### Author Response · Authors · 2025-11-21
> **Reponse**
>
> Thank you very much for the thoughtful comments. We appreciate the feedback regarding the scope of our contribution and the clarity of the method implementation. We have extensively revised the paper to address these points, adding new models, baselines, tasks, and system-level analyses.
>
> **“...new wiki dataset did not demonstrate any new experimental conclusions…”:**
>
> The two existing synthetic datasets only contain paraphrasing sets that rewrite the original text without changing important information ordering. The proposed new wiki dataset together with our synthesized paraphrases in same- and permute-orders add an important conclusion that only paraphrases permuting the original information ordering could mitigate failures in backward QA. We demonstrate that across 4 arLLMs of different sizes and model families (see more details in updated Section 4 and expanded Table 2).
>
> **Applicable Model Scope:**
>
> We expanded our evaluation from one arLLM to four arLLMs across different families and sizes (Llama-3.1-8B-Instruct, Llama-3.2-3B-Instruct, Qwen-2.5-7B-Instruct, Qwen3-4B-Instruct-2507). All the models show a consistent trend regarding our main results. We added a control baseline “reverse training” (Golovneva et. al. 2025), which performs substantially worse than dLLM and our proposed masked arLLM fine-tuning. (see updated Table 3 for more details)
>
> **Task Generalization:**
>
> To test masked fine-tuning of arLLM in broader setups, we added a new section (Section 8) on math supervised fine-tuning (SFT) with masks. We tested the fine-tuning performance of Llama-3.2-3B-Instruct and Qwen3-4B-Instruct-2507 on GSM8k and MATH, which are popular math datasets. Across all models and datasets, our proposed “masked SFT” outperforms traditional SFT. Beyond knowledge injection and reversal curse, such a result demonstrates the effectiveness of using demasking objection in arLLM fine-tuning in learning math skills. (see Section 8 for more details).
>
> **System Efficiency:**
>
> We added a new Section 9 (Computational Cost Comparison) to compare our method to other baseline methods using principled metrics like FLOPs, wall time and peak memory allocated. While our method increases per-step FLOPs (due to context length), it reaches higher peak accuracy with more than 2x convergence rate, making the overall compute still comparable to other baseline methods. Moreover, it eliminates the expensive pre-computation cost of generating high-quality paraphrases (which often involves using a capable LLM). Crucially, inference cost is identical to standard arLLMs and much lower than dLLMs. (See Section 9 for more details)
>
> **Expression Clarity:**
>
> Thanks for suggesting to clarify the exact training objective. We updated a formal formulation of masked fine-tuning (equation 2) in the updated manuscript.
>
> **“arLLMs model was trained in a pre-trained format …… Will this introduce inconsistency in comparison?”:**
>
> Thanks for this sharp catch that the training format could play a role. However, two conditions in the paper are arLLMs fine-tuned with the instruction format. One is in our ablation study that replaces the masked document in the masked arLLM prompt with random tokens (Appendix Figure 12); the other is when we use a fixed mask ratio and set the ratio to be 1 (Figure 4). In both cases, masked arLLM fine-tuning degenerated to arLLM fine-tuning with instruction formatting, and the performance of masked arLLM fine-tuning degrades to a level comparable to arLLM fine-tuning with pretraining format. These results indicate that the format is not a factor contributing to the success of masked arLLM fine-tuning.
>
> We hope our modifications address your concerns.

---

> > ### Author Response · Authors · 2025-11-26
> >
> > We sincerely thank you again for your thoughtful comments. We hope you have a chance in this busy season to evaluate our updates.
> >
> > We took the first-round review comments seriously and believe we have addressed them. If you have further concerns, please let us know. We would like to use the discussion period to address any remaining concerns.

---

### Official Review · Reviewer_aGJJ · 2025-10-30

**Soundness:** 3
**Presentation:** 2
**Contribution:** 2
**Rating:** 2
**Confidence:** 4

**Summary:**

This paper compares how arLLMs and dLLMs absorb new knowledge via fine-tuning, focusing on the reversal curse where arLLMs fail on reversed queries. Experiments on small QA datasets show arLLMs generalize poorly and depend on paraphrased data, while dLLMs are much more data-efficient, performing well on both forward and backward questions. The authors propose masked fine-tuning for arLLMs, training them to fill in masked tokens, which removes the reversal curse and closes the gap with dLLMs. Masked fine-tuned arLLMs reach near dLLM-level accuracy with similar convergence speed and no extra cost, making them highly data-efficient for knowledge injection.

**Strengths:**

1. Defines three QA datasets for evaluation.

2. Promises to release code for reproducibility.

3. Experimental setup and results are clear and easy to follow.

4. Writing is natural, structured, and easy to read.

**Weaknesses:**

1. Limited and synthetic evaluation data, hindering generalizability

The empirical validation is conducted on a very narrow set of tasks, predominantly toy or synthetic datasets. The NameDescription dataset has only 60 simple fictional statements, and the Biography dataset uses a subset of 100 short fabricated biographies, both are small and artificial. The only “real” data comes from a custom Wiki dataset of merely ~92 recent Wikipedia articles, for which QA pairs are generated automatically. Such a limited evaluation suite raises concerns about generalizability.

The method is proven only on small-scale, mostly synthetic scenarios. It is unclear if these findings would hold on larger, more diverse corpora or real-world knowledge tasks. In particular, the heavy use of GPT-generated QAs and paraphrases means the setup may not reflect the complexities of truly natural data, potentially limiting the paper’s broader relevance .

2. No testing on standard benchmarks to demonstrate broad effectiveness

The study does not evaluate the proposed masked fine-tuning approach (or the dLLM vs. arLLM comparison) on any widely-used general question-answering benchmarks (e.g. NaturalQuestions, TriviaQA or other knowledge-intensive QA tasks). All experiments are confined to the three author-curated datasets. This is a significant weakness because it weakens the claim of general effectiveness. Without results on established benchmarks, it’s hard to tell if the method would truly improve data efficiency or QA accuracy in practice. The tasks chosen are relatively constrained (mostly short factoid QAs on single paragraphs). I worry that the impressive gains might be specific to these toy tasks. Evaluating on a broader range of open-domain or knowledge-heavy benchmarks would have greatly strengthened the paper’s evidence for general applicability.

3. Unquantified overhead of masked fine-tuning

Adopting masked fine-tuning introduces additional training complexity that the paper does not adequately discuss. In each batch, the approach requires constructing masked versions of the input and potentially sampling multiple masking patterns over training. This could incur non-trivial computational overhead compared to a standard fine-tune. However, the authors do not quantify or measure this overhead.

For example, the extra runtime, memory, or complexity of generating and handling masks is never reported. They assert that masked fine-tuning is “compute-efficient”, but provide no analysis or ablation on training cost. Without such data, it remains uncertain whether the improved data efficiency might come at the expense of higher compute or implementation complexity.

A discussion or measurement of training overhead would have been helpful to confirm the method’s practical efficiency.

**Questions:**

See Weaknesses

---

> ### Author Response · Authors · 2025-11-21
> **Response**
>
> Thank you very much for a comprehensive and constructive assessment of our work.
>
> **Weakness 1: Limited and synthetic evaluation data, hindering generalizability**
>
> Thanks for pointing out the limitations of the dataset. However, fine-tuning on raw text with knowledge content and generalizing new knowledge to downstream QA ability is an unsolved open issue in LLM studies. As we discussed in the introduction and related work section, many studies demonstrated the difficulty of this task. Using a few of the wiki articles after the knowledge cut-off date is a standard setup to study this issue (Mecklenburg et. al. 2024, Ovadia et. al. 2024, and see our section 2.1 for more related works). Using LLM-generated QA and paraphrases is also well adopted in the field. Moreover, although QAs are LLM generated, we perform careful human examination to check the validity of all the QAs, which we added more details to Section 3 data construction. Similarly, the name-description and biography datasets are standards to study the reversal curse problem and are used in almost all the reversal curse papers. Considering their popularity, we chose them for the main experiments.
>
> To evaluate the generalizability of our proposed “masked fine-tuning” method, we added a new math supervised fine-tuning (SFT) section (Section 8). We fine-tune Llama-3.2-3B-Instruct and Qwen3-4B-Instruct-2507 on GSM8k and MATH dataset. SFT with our proposed masked fine-tuning objective outperforms the traditional SFT method. This suggests that the data efficiency benefits of the masked ar fine-tuning objective extend not just to knowledge tasks/factual recall, but to general instruction following and chain-of-thought learning reasoning. Please see the newly added section 8 for more details.
>
> **Weakness 2: No testing on standard benchmarks to demonstrate broad effectiveness**
>
> We appreciate the request to demonstrate broad effectiveness and we agree; thus, beyond knowledge injection/reversal curse, we added an SFT experiment with a demasking objective on two popular math datasets (Section 8), and also extended the number of testing models (updated Table 1).  In the math SFT task, we indeed found that across all models and datasets, our proposed masked SFT outperforms traditional SFT (details in Section 8).
>
> However, there is no popular benchmark for evaluating knowledge fine-tuning. The NaturalQuestions and TriviaQA benchmarks you mentioned are not suitable for testing knowledge injection. They are used for evaluating the general knowledge of a pre-trained model, i.e. they do not offer a knowledge document for each question-answer pair, and the related knowledge is supposed to be general and learned during pre-training, rather than new knowledge learned during fine-tuning. This is also the reason why researchers often use a synthetic dataset for studying knowledge injection: the pre-trained model does not know synthesized fictitious knowledge. For it to be a true knowledge injection task, it is crucial that the studied datasets are completely novel to the model (in the sense of pretraining dataset contamination), as we demonstrate on our datasets through the model’s low “before-fine-tuning” accuracy.
>
> **Weakness 3: Unquantified overhead of masked fine-tuning**
>
> Thanks for pointing this out. Consequently, we added a comprehensive section on the analysis of the compute and memory complexity (see newly added Section 9). Briefly, for your concern about the cost of sampling the random mask, the sampling and prompt preparation time is negligible (<3.5ms/step) compared to the wall time of computing the forward and backward path (~2.5s/step), similar for the memory cost in the sampling process (details in Section 9). The main additional cost of “masked fine-tuning” per training step is from the increased prompt length caused by reformatting the original text as SFT style. However, our method reaches higher peak accuracy with more than 2x convergence rate, making the overall compute still comparable to other baseline methods. Our main claim on the “data-efficiency” is defined as how much fine-tuning data (including paraphrases) is required for a pre-trained model to generalize the knowledge in the data to downstream tasks such as QA. We claim our method to be data efficient as it reaches high forward and backward QA accuracy without relying on any additional LLM-generated paraphrases. We make this point clearer in the updated manuscript.
>
> We hope these quantitative additions regarding the robustness of our method and a computational cost analysis address your comments.

---

> > ### Comment · Reviewer_aGJJ · 2025-11-24
> >
> > Having read the rebuttal and revised manuscript, I think the authors have substantially addressed my main concerns, though some scope limitations remain and should be stated clearly.
> >
> > On data and benchmarks, the knowledge-injection experiments still rely on relatively small, synthetic or controlled datasets, but these are in line with prior work on the reversal curse. The added GSM8K and MATH results show that masked SFT consistently outperforms vanilla SFT on standard math reasoning benchmarks, suggesting the objective is not simply overfitting to toy setups. That said, we still lack evidence in large-scale, heterogeneous, real-world knowledge scenarios, so the claims should be framed accordingly.
> >
> > Regarding the absence of NQ/TriviaQA, the authors make a reasonable case that such benchmarks are ill-suited for “true knowledge injection,” since they test pre-existing knowledge rather than newly introduced, document-specific information. Combined with the broader model set and added math experiments, my concern has been reduced to one of breadth rather than validity.
> >
> > My worry about unquantified training overhead is fully resolved: the new analysis shows that mask sampling and prompt construction contribute negligible cost compared to forward/backward passes, and although per-step FLOPs increase, convergence is faster and final accuracy is higher.
> >
> > Overall, the paper now presents a convincing and conceptually clean masked fine-tuning objective for arLLMs that recovers much of the data efficiency of dLLMs without architectural changes. The work is empirically solid within its studied regime, with remaining limitations mainly about generality to more realistic, large-scale knowledge-intensive settings. Therefore, I change my score to 4.

---

> ### Author Response · Authors · 2025-11-26
>
> Thank you very much for raising the score. We sincerely appreciate your constructive feedback and are glad that the additional experiments addressed your concerns.

---

### Official Review · Reviewer_ex6w · 2025-10-31

**Soundness:** 3
**Presentation:** 3
**Contribution:** 3
**Rating:** 6
**Confidence:** 2

**Summary:**

The authors look at the reversal-curse problems in Autoregressive LLMS (AR-LLMs) and Diffusion LLMs (D-LLMs) and investigate D-LLMs' capabilities in those lacking areas. Further, the authors propose a masked paradigm for AR-LLMs mimicking masked diffusion to bridge the gap between AR-LLMs and D-LLMs.

## Impact of paraphrasing

1. Re-finding previous results, the authors show that the backward accuracy in AR-LLMs is close to zero with next token prediction, but paraphrasing-based data increases the accuracy to 1.
2. On the other hand, D-LLMs perform well (slightly worse than AR-LLMs ) on the fwd tasks and better on the bwd tasks.
3. "his indicates that dLLM does not trade better data efficiency and performance for more
computations; it requires the same or less computation and fewer training samples, but achieves
better downstream performance."

## Masked fine-tuning

1. The authors propose masked fine-tuning, mimicking the D-LLMs training paradigm where the objective is to predict the masked intermediate tokens, dropped out at random.
2. Authors explore the effect of mask-ratio, and Figure 4 indicates ratio=0.75 (with paraphrases) / raio=0.5 w/o paraphrases performs the best for AR-LLMs.

**Strengths:**

1. The authors present a simple, clear empirical report and training modifications to look at the reversal curse in AR and D LLMs.
2. The contributions look novel and have strong empirical results.

**Weaknesses:**

1. Efficiency seems poorly defined - taking a principled approach (like FLOPs, etc.) and showing concrete metrics to show efficiency-based claims.
2. deep dive around paraphases and ratio seems missing since behavior is not consistent and it is not obvious why.

**Questions:**

Just the weakness above.

---

> ### Author Response · Authors · 2025-11-21
> **Response**
>
> Thank you very much for recognizing the novelty and empirical strength of our study. You may read our common message for a summary of all the revision updates. Regarding your comments:
>
> **1.“Efficiency seems poorly defined”**
>
> Thanks for pointing out the unclarity. In the updated manuscript, we make its definition clearer in the introduction. In our study, we use “data-efficiency” to specifically mean how much fine-tuning data (including paraphrases) is required for a pre-trained model to generalize the knowledge in the data to downstream tasks such as QA. We claim our method to be data efficient as it reaches high forward and backward QA accuracy without relying on LLM-generated paraphrases. Beyond data-efficiency, we also added a section (Section 9) on systemic analysis of the compute- and memory-efficiency using principled metrics like FLOPs, wall time and peak memory allocated. We would like to clarify that our main claim is not our method is more compute- and memory-efficient, in fact, our method costs more compute compared to baseline methods per training step. However, our method reaches higher peak accuracy with more than 2x convergence rate, making the overall compute comparable.
>
>
> **2. “deep dive around paraphrases and ratio seems missing”**
>
> Thanks for the suggestion. In the Appendix section “On reversal curse”, we added more discussion on the role of paraphrases in nursing the reversal curse. And in the newly added Section 9, we discuss the cost of adding paraphrases. In updated Table 1, we expand previous results and show that same order paraphrases consistently improve forward QA performance but not backward QA across 4 models and 3 datasets. This effect indicates that for standard arLLM fine-tuning, key information ordering in the paraphrased texts greatly affects downstream QA performance.
>
> For the effect of “mask ratio”, we added the following discussion in the main text: “Conceptually, varying $t$ modulates task difficulty; a moderate $t$ places training in a “hard-but-not-impossible” regime that maximizes the per-example gradient signal. The effectiveness of some fixed mask ratios indicate that a dLLM only needs to sample over t to learn the demasking process during pre-training. Once this ability is acquired and not forgotten during fine-tuning, a fixed $t$ is sufficient for learning new knowledge.”
>
> We hope our modifications address your concerns.

---

> > ### Author Response · Authors · 2025-11-26
> >
> > We sincerely thank you again for your thoughtful comments. We hope you have a chance in this busy season to evaluate our updates.
> >
> > We took the first-round review comments seriously and believe we have addressed them. If you have further concerns, please let us know. We would like to use the discussion period to address any remaining concerns.

---

### Official Review · Reviewer_9H3S · 2025-11-03

**Soundness:** 2
**Presentation:** 3
**Contribution:** 2
**Rating:** 4
**Confidence:** 4

**Summary:**

Building on prior findings that masked diffusion language models (dLLMs) exhibit strong data efficiency and are free from the “reversal curse” during pretraining, this paper investigates whether these advantages also hold in the post-training stage. The study first demonstrates that dLLMs maintain data efficiency in both forward and backward question answering (QA) tasks without requiring paraphrases, whereas autoregressive language models (arLLMs) heavily depend on paraphrasing to generalize textual knowledge into QA performance. Motivated by this observation, the paper further proposes a masked fine-tuning method for effective knowledge injection in arLLMs.

**Strengths:**

1.	It is interesting and novel to explore diffusion language models (dLLMs) in the context of the post-training stage, demonstrating their data efficiency for knowledge injection in QA tasks.
2.	The proposed masked fine-tuning approach for autoregressive LLMs (arLLMs) is well motivated by the empirical findings that dLLMs exhibit strong performance on QA tasks, and the method itself is both interesting and effective.
3.	The presentation is generally clear and well organized, with detailed descriptions that make the paper easy to follow.

**Weaknesses:**

1.	The proposed masked fine-tuning approach is simple and effective; however, it is not entirely convincing whether it offers substantial improvements over existing fine-tuning methods. More extensive comparisons with prior approaches are needed. For instance, the following work may be relevant:
https://arxiv.org/pdf/2403.13799.
Additionally, other simple data augmentation strategies that could address similar issues in autoregressive LLMs should be compared.
2.	The proposed masked fine-tuning method appears conceptually related to T5’s infilling objective, and it is unclear whether its technical novelty is significant relative to other similar objective functions.
3.	The paper attributes the data efficiency of dLLMs to their bidirectionality, but such characteristics are not unique to diffusion models. The motivation for the current approach could have been derived equally well from BERT-like bidirectional models, without relying specifically on dLLMs. As a result, the conceptual connection between dLLMs and the proposed method feels somewhat weak.

**Questions:**

1. Can the masked fine-tuning be extended to other general NLP tasks?

---

> ### Author Response · Authors · 2025-11-21
> **Response**
>
> Thank you very much for the constructive comments. You may read our common message for a summary of all the revision updates. Here is our response to your comments.
>
> **Weakness 1. Comparing existing fine-tuning methods**
>
> Thanks for your suggestion on adding “reverse training” in Golovneva et al., 2024 as a baseline control. We updated Table 1 to include its performance, and expanded experiments to four different arLLM models. The “reverse training” baseline only slightly improves both forward and backward QA, and falls far behind masked AR (our method).
>
> This reverse training method together with the same order paraphrasing for our two synthetic datasets (NameDescription and Biography) are popular “simple” data augmentation strategies as they don’t require LLM-based rewriting. Our updated Table 1 shows that across four different arLLM models our method significantly outperforms these augmentation strategies in solving the reversal curse.
>
> **Weakness 2. “Connections to T5”.**
>
> Thanks for your insight. Indeed, our proposed masked AR fine-tuning method is closely related to T5 and BERT denoising objectives. However, our study contributes differently from those prior studies in several ways. 1) Adapting the objective effectively in arLLM post-training is novel. Architecturally, T5 is encoder-decoder; BERT is encoder-only, whereas most arLLMs (Llama, Qwen etc.) are decoder-only. As decoder-only arLLMs are becoming the dominating paradigm, our study aims to demonstrate a method that requires minimal architectural or objective changes to decoder-only arLLMs and can be applied in the post-training stage to overcome their inherent pitfall (e.g., reversal curse) compared to other paradigms. 2) Applying this post-training recipe to solve arLLM knowledge injection failure is novel. Knowledge injection by finetuning consistently lags behind RAG (Ovadia & Brief et. al., 2024) and relies on extensive paraphrases. Our simple recipe improves arLLM knowledge injection w/o paraphrases accuracy (forward & backward QA) from almost 0 to ~0.9, offering significant practical implications in developing LLMs as lifelong learners.
>
> We updated our manuscript with the above discussion.
>
> **Weakness 3. “Bert-like models”.**
>
> “The motivation for the current approach could have been derived equally well from BERT-like bidirectional models.” We chose to motivate our work with the dLLM because there has been evidence (Nie et. al. 2025) showing pre-trained dLLM does not suffer from the reversal curse, but on the other hand, BERT models have very poor performance on both forward and reverse QA tasks (“Physics of Language Models: Part 3.1” Zhu et. al. 2024, “The Factorization Curse” Kitouni et. 2024), because of BERT as a non-generative model struggles with generating long high-quality answers. As noted by Nie et al., changing the BERT training objective with sampling mask ratio between 0 and 1 can turn it into a principled generative model with decent instruction following and QA ability, which is now known as masked diffusion models. This also becomes our motivation to test the effect of “fixed mask ratio” in section 7 which makes the objective more similar to BERT models.
>
> We added more discussion on this point in the related works section.
>
> **Question 1. “other general NLP tasks”**
>
> Thanks for suggesting. We agree, and in the revised paper, we expanded our evaluation beyond Knowledge Injection to math tasks. On two different arLLMs and two math datasets (GSM8K and MATH), our masked fine-tuning paradigm outperforms standard SFT. This suggests that the benefits of the masked ar fine-tuning objective extend not just to knowledge tasks/factual recall, but to general instruction following and chain-of-thought learning. Please see new section 8 for more details.
>
> We hope our modifications address your concerns.

---

> > ### Author Response · Authors · 2025-11-26
> >
> > We sincerely thank you again for your thoughtful comments. We hope you have a chance in this busy season to evaluate our updates.
> >
> > We took the first-round review comments seriously and believe we have addressed them. If you have further concerns, please let us know. We would like to use the discussion period to address any remaining concerns.

---

### Author Response · Authors · 2025-11-21
**Revision Summary**

Dear Reviewers and AC,

We sincerely thank all reviewers for taking the time to provide constructive feedback. We appreciate that you found our study novel, having strong empirical results, and easy to read. We also value the critiques regarding our limited experimental setup and the lack of system-level analysis (e.g., compute overhead). In response, we added substantial supporting experiments and new sections presenting these results and the system-level analysis. Below is a brief summary of the updates:

* We added a suggested baseline control method, “reverse training” (Golovneva et al., 2024), which claims to address the reversal curse; our method significantly outperforms it (Table 1).
* We tested our masked fine-tuning paradigm on three additional arLLMs (Llama-3.2-3B-Instruct, Qwen2.5-7B-Instruct and Qwen3-4B-Instruct-2507), bringing the total to four arLLMs. All models show a consistent trend regarding our main results.
* Moving beyond “knowledge injection” and QA tasks, we evaluated our masked fine-tuning paradigm via SFT on math datasets (GSM8K and MATH); our masking method has higher accuracy than traditional SFT across all models and both datasets (see new Section 8)
* We added a section presenting a system-level analysis (i.e. FLOPs, memory, wall time) comparing our method against other baseline methods (see new Section 9)
* We added a more formal description of our masked fine-tuning paradigm (see eq. 2)

Beyond these updates, we have further polished the writing in line with your comments. Please see the detailed responses in separate messages.

---

### Author Response · Authors · 2025-11-26

Dear AC,

We took the first-round review comments seriously and believe we have addressed all of them (see "Revision Summary"). However, we only heard back from 1 of 4 reviewers before the sudden closure of the discussion. One reviewer raised the score from 2 to 4; the other three are 4, 4, 6 without responding to our rebuttal.

We believe our paper is novel and significant on the important topic of how to effectively inject new knowledge into the model using limited data in the post-training phase. We studied this problem in various setups and developed a method that outperforms the traditional fine-tuning by a large margin in this regime. The manuscript would interest a broad audience who are interested in LLM factual memory, life-long learning, data-efficient fine-tuning, etc.

Thanks.

---

### Meta-Review · Area_Chair_MR63 · 2026-01-02

**Summary:**

The paper investigates the reversal curse and data efficiency in knowledge injection for arLLMs compared to dLLMs. The authors demonstrate that while arLLMs struggle with reversed queries without extensive paraphrasing, dLLMs are robust. Inspired by this, the paper proposes a masked fine-tuning paradigm for arLLMs, where the model is prompted to recover a masked version of the text, which is claimed to close the performance gap with dLLMs. The authors also extend this method to mathematical SFT.

**Reviewer Concerns:**

The reviewers generally appreciated the clarity of the problem statement and the interesting empirical analysis comparing arLLMs and dLLMs in the post-training phase. The proposed method is simple and effective within the experimental setup provided. The authors engaged actively during the rebuttal, adding a reverse training baseline, expanding the model suite to four arLLMs, and including experiments on Math SFT (GSM8K and MATH) to demonstrate broader applicability.

**Reviewer Scores:**

- Reviewer aGJJ & haT6 shared concerns around the synthetic or small-scale datasets. While the authors argue that synthetic data is standard for "reversal curse" studies, the claim of general "knowledge injection" remains under-supported without evaluation on more realistic, large-scale, or heterogeneous benchmarks. The addition of Math SFT experiments helps, but it shifts the domain rather than fully validating the core claim regarding factual knowledge injection in realistic settings.
- Reviewer 9H3S & haT6 shared concerns around the technical novelty and contribution significance. The proposed "masked fine-tuning" is somewhat similar to established denoising objectives (like T5's infilling). While applying this to decoder-only models for the specific purpose of mitigating the reversal curse is an interesting application, reviewers felt the technical leap was incremental. The connection to dLLMs serves as a motivation but does not necessarily distinguish the method from simply applying bidirectional-style objectives to arLLMs.

---

### Decision · Program_Chairs · 2026-01-26

Reject